# Internal amino acid state modulates yeast taste neurons to support protein homeostasis in *Drosophila*

Kathrin Steck[†], Samuel J Walker[†], Pavel M Itskov, Célia Baltazar, José-Maria Moreira, Carlos Ribeiro*

Champalimaud Centre for the Unknown, Lisbon, Portugal

**Abstract** To optimize fitness, animals must dynamically match food choices to their current needs. For drosophilids, yeast fulfills most dietary protein and micronutrient requirements. While several yeast metabolites activate known gustatory receptor neurons (GRNs) in *Drosophila melanogaster*, the chemosensory channels mediating yeast feeding remain unknown. Here we identify a class of proboscis GRNs required for yeast intake. Within this class, taste peg GRNs are specifically required to sustain yeast feeding. Sensillar GRNs, however, mediate feeding initiation. Furthermore, the response of yeast GRNs, but not sweet GRNs, is enhanced following deprivation from amino acids, providing a potential basis for protein-specific appetite. Although nutritional and reproductive states synergistically increase yeast appetite, reproductive state acts independently of nutritional state, modulating processing downstream of GRNs. Together, these results suggest that different internal states act at distinct levels of a dedicated gustatory circuit to elicit nutrient-specific appetites towards a complex, ecologically relevant protein source.
DOI: https://doi.org/10.7554/eLife.31625.001

*For correspondence:
carlos.ribeiro@neuro.
fchampalimaud.org

[†]These authors contributed equally to this work

## Introduction

Decision-making is a key function of the brain. One of the most ancestral and consequential decisions animals need to make is which foods to eat, since balancing the intake of multiple classes of nutrients is critical to optimizing lifespan and reproduction (*Simpson and Raubenheimer, 2012*). To do this, many animals, including humans, develop so-called 'specific appetites', seeking out and consuming specific foods in response to a physiological deficit of a particular nutrient (*Beauchamp et al., 1990*; *Deutsch et al., 1989*; *Hughes and Wood-Gush, 1971*; *Itskov and Ribeiro, 2013*). Recently, several populations of central neurons driving consumption of specific nutrients have been identified in different species (*Jarvie and Palmiter, 2017*; *Liu et al., 2017*; *Matsuda et al., 2017*; *Sun et al., 2017*). How these circuits modulate sensory processing to elicit state-specific behavioral responses, however, is poorly understood.

The ability to precisely control the intake of dietary proteins is emerging as a conserved phenomenon across phyla. Insects, for example, tightly regulate their intake of protein depending on their internal states (*Dethier, 1961*; *Simpson and Abisgold, 1985*). Mosquito disease vectors impose a huge burden on human health due to their need for dietary protein, which drives host-seeking and feeding behaviors only during specific internal states (*Brown et al., 1994*; *Judson, 1967*). Dietary protein homeostasis is not specific to invertebrates, as humans are also able to select high-protein foods when low on protein (*Gosby et al., 2011*; *Griffioen-Roose et al., 2012*). Although protein is essential for sustaining key physiological processes such as reproduction, excessive protein intake has detrimental effects on aging and health (*Grandison et al., 2009*; *Lee et al., 2008*; *Levine et al., 2014*; *Piper et al., 2014*; *Solon-Biet et al., 2014*). This emphasizes the importance of this tight control of protein intake.

**eLife digest**   When animals run low on a certain nutrient, they change their behavior to seek out and feed on foods rich in that missing element. For example, fruit flies lacking sugar will look for and eat more sweet food; if they need proteins, they will instead favor yeast, flies' principal source of proteins.

In fruit flies, certain neurons on the insects' tongue (or proboscis) are dedicated only to taste. These cells are divided in groups specialized for a type of nutrient – for instance some of them only react to sugar. Taste neurons sense food and help coordinate how much and for how long the animals will feed. However, despite how important proteins are for flies, the neurons dedicated to tasting yeast had yet to be identified.

Here, Steck, Walker et al. report discovering a new set of taste neurons in fruit flies, which are activated by a unique combination of molecules present in yeast. Crucially, without these neurons being active, the insects can no longer adjust their diet to eat more yeast when they are deprived of proteins. The activity of these cells is also regulated by internal levels of nutrients derived from proteins.

The yeast-specific taste neurons are present in two areas on the fly's proboscis, which is used like a straw when feeding. The two sets of cells have different roles in the consumption of yeast. The first group, which is located at the extremity of the proboscis, helps flies detect and start consuming the resource. The second group, which is on the inner surface of the proboscis, influences whether the insects keep feeding. If one of these groups of neurons is deactivated, flies continue to eat yeast as normal, showing that the system is redundant. However, if both sets are turned off artificially, the insects stop favoring yeast even when they are in need of proteins.

Steck, Walker et al. show how the animals' internal states also influence the activity of these neurons. When the insects are deprived of molecules that are only found in proteins, these newly discovered neurons are primed to react more strongly when they are exposed to yeast. This potentially makes flies eat more yeast, and as a result consume more proteins.

Many biological systems in flies are similar in other insects and even humans. If this is the case for these taste neurons, fruit flies could be a good model to study how pests such as locusts and mosquitoes are attracted to the proteins in crops and blood, but also how humans make decisions about food.

DOI: https://doi.org/10.7554/eLife.31625.002

Most *Drosophila* species, including the model organism *Drosophila melanogaster*, are highly adapted to consume yeast as the major source of non-caloric nutrients in the wild, including proteins, and thus amino acids (AAs) (*Baumberger, 1919*; *Camargo and Phaff, 1957*; *Phaff et al., 1956*), as well as other nutrients such as sterols and vitamins (*Starmer and Lachance, 2011*). It is therefore essential for flies to precisely regulate the intake of yeast. This is achieved by modulating decision-making at different scales, from exploration to feeding microstructure, according to different internal state signals, including AA state, reproductive state, and commensal bacteria composition (*Corrales-Carvajal et al., 2016*; *Leitão-Gonçalves et al., 2017*). Deprivation from dietary protein or essential AAs leads to a compensatory yeast appetite, thought to be mediated by direct neuronal nutrient sensing (*Leitão-Gonçalves et al., 2017*; *Piper et al., 2014*; *Ribeiro and Dickson, 2010*; *Vargas et al., 2010*). Activation of a small cluster of dopaminergic neurons by AA deficit is involved in stimulating this appetite (*Liu et al., 2017*); while in parallel, a protein-specific satiety hormone, FIT, is secreted by the fat body in the fed state and inhibits intake of protein-rich food (*Sun et al., 2017*). Mating further enhances the yeast appetite of protein-deprived females through the action of Sex Peptide on a dedicated neuronal circuit (*Walker et al., 2015*, *2017*). Finally, specific commensal bacteria can suppress the yeast appetite generated by deprivation from essential amino acids, through a mechanism likely to be distinct from simply providing the missing amino acids (*Leitão-Gonçalves et al., 2017*). How the nervous system integrates these different internal state signals, and how these signals modulate neural circuits that process chemosensory information and control feeding behavior, is unknown.

Contact chemosensation is critical for assessing the value of food sources for feeding (*Dethier, 1976*; *Yarmolinsky et al., 2009*), as well as for a range of other behaviors (*Auer and Benton, 2016*; *Hussain et al., 2016b*; *Joseph et al., 2009*; *Joseph and Heberlein, 2012*; *Mann et al., 2013*; *Thoma et al., 2016*; *Yang et al., 2008*, *2015*). As in many insects, *Drosophila* gustatory receptor neurons (GRNs) are distributed in different parts of the body (*Stocker, 1994*), and their central projections are segregated according to the organ of origin (*Wang et al., 2004*). Furthermore, different classes of GRNs are thought to be specialized to detect different categories of taste stimuli, including bitter compounds, sugars, water and sodium (*Cameron et al., 2010*; *Du et al., 2016*; *Fujishiro et al., 1984*; *Inoshita and Tanimura, 2006*; *Marella et al., 2006*; *Miyamoto et al., 2013*; *Soldano et al., 2016*; *Wang et al., 2004*; *Weiss et al., 2011*; *Zhang et al., 2013*). Of these, the most well-characterized are sweet-sensing GRNs, which mediate attractive responses to sugars (*Fujishiro et al., 1984*; *Marella et al., 2006*; *Miyamoto et al., 2013*; *Wang et al., 2004*), fatty acids (*Masek and Keene, 2013*; *Tauber et al., 2017*) and glycerol (*Wisotsky et al., 2011*). Recent work has demonstrated that sweet GRNs innervating labellar taste sensilla and pharyngeal taste organs act in parallel to support sugar feeding, since loss of taste sensilla alone does not abolish flies' feeding preference for sugar, but combining this with silencing of pharyngeal sugar-sensing neurons drastically reduces sugar preference (*LeDue et al., 2015*). Specific roles have been demonstrated for these distinct sweet-sensing GRN populations: sensillar GRNs are important for initiation of feeding, suppression of locomotion (*Mann et al., 2013*; *Thoma et al., 2016*), and inhibition of egg-laying (*Yang et al., 2008*, *2015*), while pharyngeal GRNs are important for sustaining sugar feeding (*LeDue et al., 2015*; *Yapici et al., 2016*), limiting sucrose intake (*Joseph et al., 2017*), and inducing local search behavior (*Murata et al., 2017*). These different classes of neurons project to discrete regions within the subesophageal zone (SEZ) in the brain of the adult fly (*Wang et al., 2004*), providing a potential substrate for encoding different gustatory categories. However, unlike sensillar and pharyngeal GRNs, the role of GRNs innervating taste pegs in feeding behavior remains unknown. Moreover, the involvement of the gustatory system in yeast feeding is currently unclear.

In order to dissect the neural circuit mechanisms by which internal states homeostatically regulate protein intake, it is crucial to identify the sensory inputs that drive intake of yeast. While multiple volatiles produced by yeast fermentation are detected by the olfactory system and are highly attractive at long ranges to fruit flies (*Becher et al., 2012*; *Christiaens et al., 2014*; *Dweck et al., 2015*), the sensory channels that mediate feeding on yeast are unknown. Several yeast metabolites have been shown to activate subsets of taste receptor neurons in *Drosophila*. Sensillar GRNs detect glycerol, a sugar alcohol produced by yeast, through the Gr64e receptor (*Wisotsky et al., 2011*), while some amino acids have been shown to activate a subset of *Ir76b*-expressing GRNs on the legs (*Ganguly et al., 2017*). Taste peg GRNs, meanwhile, have been shown to respond to carbonation, a major byproduct of alcoholic fermentation (*Fischler et al., 2007*), although whether this contributes to feeding has never been tested. Whether and to what extent these individual gustatory cues contribute to the high phagostimulatory power of yeast, however, is currently unknown; and how sensory neurons in different anatomical locations coordinately support yeast feeding is not understood. Thus, a major gap in our understanding of the neuroethology of *Drosophila* is that the specific sensory inputs that drive feeding on yeast remain to be characterized.

In this study, we show that a subset of GRNs on the proboscis is required for yeast feeding behavior. Acute silencing of these neurons drastically reduces feeding on yeast. We show that within this population of GRNs, neurons in distinct anatomical locations show physiological responses to the taste of yeast, and that while these subsets control distinct aspects of yeast feeding behavior, they are ultimately redundant in terms of total yeast feeding. Specifically, sensillar GRNs are essential for feeding initiation by proboscis extension in response to yeast taste. Moreover, using closed-loop optogenetic silencing, we demonstrate that GRNs innervating taste pegs are essential for sustaining yeast feeding bursts after they have been initiated. We further show that the responses of both sensillar and taste peg sensory neurons to yeast are modulated by the internal AA state of the fly: deprivation from dietary AAs increases both yeast feeding and the gain of sensory neuron responses. This effect is specific to yeast GRNs, as sweet-sensing neurons are not sensitized by yeast deprivation. Furthermore, while reproductive state modulates yeast feeding behavior, it has no effect on sensory responses, indicating that distinct internal states act at different levels of sensory processing to modulate behavior. This study therefore identifies gustatory neurons that are required

for the ingestion of an ethologically and ecologically key resource of the fly and describes a circuit mechanism that could contribute to the homeostatic regulation of protein intake.

## Results

### Identification of sensory neurons underlying yeast feeding

Yeast is a key nutrient source in the ecology of *Drosophila* species (*Camargo and Phaff, 1957*; *Phaff et al., 1956*). While sugars provide flies with energy, yeast is the primary source of nutrients beyond just calories for the adult fly, and particularly of amino acids (AAs) and proteins. As such, flies can independently regulate their intake of sugars and yeast depending on their internal state in order to compensate for nutritional deficiencies. In mated females, deprivation from AAs specifically increases feeding on yeast, whereas deprivation from dietary carbohydrates elicits a specific increase in feeding on sucrose (*Figure 1A*). Though much is known about the gustatory pathways by which flies regulate intake of sugars (*Fujishiro et al., 1984*; *Inagaki et al., 2012*; *Jiao et al., 2008*; *LeDue et al., 2015*; *Marella et al., 2006*, *Marella et al., 2012*; *Miyamoto et al., 2013*; *Wang et al., 2004*; *Yapici et al., 2016*), the sensory basis of yeast feeding is not known.

Several yeast metabolites, such as glycerol, carbonation and polyamines, have been shown to activate chemosensory neurons of *Drosophila* (*Fischler et al., 2007*; *Ganguly et al., 2017*; *Hussain et al., 2016b*; *Wisotsky et al., 2011*). As such, we first asked whether these metabolites can account for the high phagostimulatory power of yeast in protein-deprived mated females. In these experiments, we deprived mated females of protein for 10 days, in order to evoke a high level of motivation to feed on yeast. We found that while yeast induced an extremely high level of feeding, these other substrates evoked only a very low level of feeding (*Figure 1—figure supplement 1A*). We also found that deactivating yeast, such that it cannot generate carbonation, did not affect the high level of feeding on this substrate. Furthermore, silencing gustatory neurons known to respond to carbonation and glycerol did not affect yeast preference, reinforcing the idea that individually, these specific yeast metabolites do not explain the high phagostimulatory power of yeast (*Figure 1—figure supplement 1B and C*). For these reasons, we devised a strategy to identify neurons required for feeding on yeast, flies' natural food source.

We set out to isolate neurons which, when their synaptic output is acutely blocked, would abolish the high feeding preference of protein-deprived mated females for yeast (*Figure 1—figure supplement 2A*). To this end, we employed a two-color food choice assay in which protein-deprived flies had the choice of eating sucrose or yeast (*Ribeiro and Dickson, 2010*) to conduct an unbiased silencing screen of enhancer-trap lines (*Figure 1—figure supplement 2B and C*). In this assay, the number of flies feeding on yeast, sucrose or both was visually scored based on abdomen color, and yeast preference calculated as $(n_{yeast} - n_{sucrose})/n_{total}$. Using this approach, we identified one line (*1261-GAL4*) which showed a strong reduction in yeast preference compared to genetic and temperature controls (*Figure 1B*) and labeled subsets of chemosensory neurons in the head and legs (*Figure 1C*). As such, we designed a targeted follow-up screen in which we silenced subsets of chemosensory and neuromodulatory neurons (*Supplementary file 1*) and recovered two more lines that reduced the yeast preference of protein-deprived females when their output was acutely blocked during food choice: *Ir76b-GAL4* and *Ir25a-GAL4* (*Figure 1D*). As with *1261-GAL4*, these lines labeled subsets of chemosensory neurons in the head and legs (*Figure 1—figure supplement 2D–F*).

The loss of yeast preference upon silencing the neurons labeled in these lines could be due to a reduction in yeast feeding, and/or to an increase in sucrose feeding. To disambiguate these possibilities, we used the ability provided by the flyPAD to monitor the feeding of single flies on individual substrates with a resolution of single sips (*Itskov et al., 2014*). We found that in protein-deprived flies, silencing of the neurons labeled in any of thesese *GAL4* lines specifically reduced yeast feeding, without increasing sucrose feeding, confirming that these neurons are required for feeding on yeast (*Figure 1—figure supplement 2G*). To further test the specificity of this manipulation to yeast, we deprived flies from either AAs or carbohydrates, using a holidic diet, and subsequently measured feeding on the flyPAD. Silencing *1261-GAL4* reduced yeast feeding in AA-deprived flies, but did not reduce sucrose feeding in carbohydrate-deprived flies, further highlighting the specificity of this phenotype (*Figure 1—figure supplement 2H*).

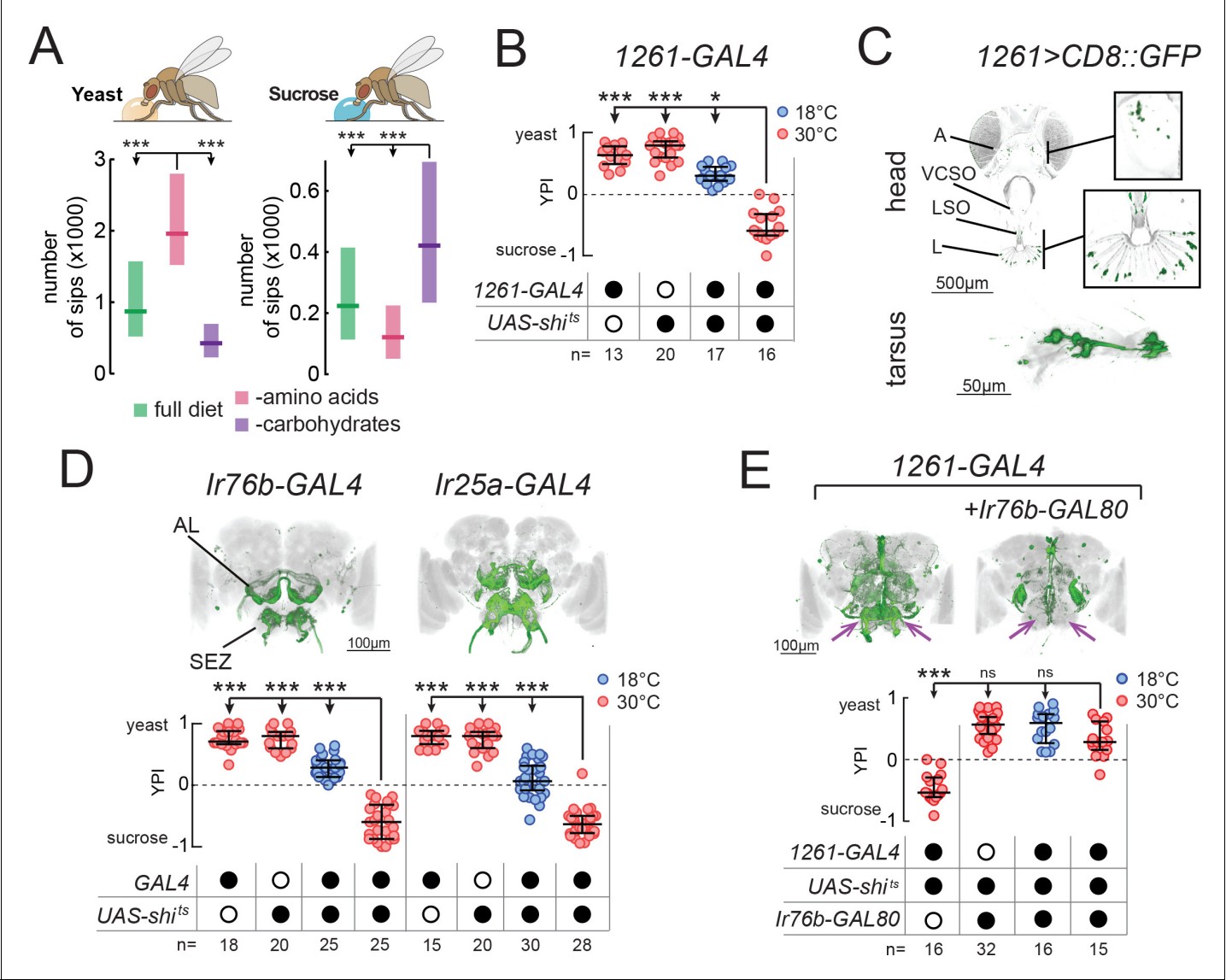

**Figure 1.** Identification of neuronal populations required for yeast feeding. (A) Number of sips from 10% yeast (left) and 20 mM sucrose (right) by mated female flies of the genotype *Ir76b-GAL4,UAS-GCaMP6s*, fed for 3 days on a holidic diet with the indicated composition. (B) Yeast preference index (YPI) of yeast-deprived female flies in which *1261-GAL4* neurons were acutely silenced and corresponding controls. (C) Expression pattern of *1261-GAL4* in the head and tarsus of the fly. Boxes are magnifications of the antenna or the labellum. Note absence of signal in the maxillary palps. Green represents GFP signal and gray the cuticular autofluorescence. (D) YPI of yeast-deprived female flies in which *Ir76b-* or *Ir25a-GAL4* neurons were acutely silenced and corresponding controls. (E) YPI of yeast-deprived female flies in which *1261-GAL4* neurons were all silenced, or with *Ir76b-GAL80*, and corresponding controls. Arrows indicate presence/loss of GFP expression in the SEZ. (D) and (E) Expression pattern of experimental flies in the brain as visualized using *UAS-CD8::GFP* in green, with nc82 synaptic staining in gray. Note absence of cell bodies in the *Ir25a-GAL4* brain. A, antenna; VCSO, ventral cibarial sense organ; LSO, labral sense organ; L, labellum; AL, antennal lobes; SEZ, subesophageal zone. In this and following figures, empty and filled black circles represent absence and presence of the indicated elements, respectively. In (A), boxes represent median with upper/lower quartiles. In (B), (D) and (E), circles represent yeast preference in single assays, with line representing the median and whiskers the interquartile range. \*\*\*p<0.001, \*p<0.05, ns p≥0.05. Groups compared by Kruskal-Wallis test, followed by Dunn's multiple comparison test.

DOI: https://doi.org/10.7554/eLife.31625.003

The following figure supplements are available for figure 1:

**Figure supplement 1.** Individual yeast metabolites do not explain the phagostimulatory power of yeast.
DOI: https://doi.org/10.7554/eLife.31625.004

**Figure supplement 2.** Characterization of neuronal populations required for yeast feeding.
DOI: https://doi.org/10.7554/eLife.31625.005

*Figure 1 continued on next page*

*Figure 1 continued*

**Figure supplement 3.** The idenitified lines label neurons required for amino acid preference and amino acids are not the sole determinant of yeast feeding.

DOI: https://doi.org/10.7554/eLife.31625.006

## *1261-, Ir76b-* and *Ir25a-GAL4* label common neurons required for yeast feeding

All of these lines drive expression in both olfactory (ORNs) and gustatory (GRNs) receptor neurons (*Figure 1C* and *Figure 1—figure supplement 2D–F*). ORNs were labeled in the antennae, but not the maxillary palps (A, *Figure 1C*; and *Figure 1—figure supplement 2D and E*), and projected to multiple non-overlapping sets of glomeruli of the antennal lobes (AL, *Figure 1D and E*). GRNs were labeled on the labellum (L) and in pharyngeal taste organs (LSO, VCSO), as well as on the legs (*Figure 1C*, *Figure 1—figure supplement 2D and E*, *Supplementary file 1*). *1261-GAL4* exhibited more sparse expression in GRNs than the other lines, including in pharyngeal GRNs (*Supplementary file 1*). GRN projections could be seen in the subesophageal zone (SEZ, *Figure 1D and E*), as well as in leg and wing neuropils of the ventral nerve cord (VNC, FL, ML, HL, W, *Figure 1—figure supplement 2F*).

Since all of these GAL4 lines label chemosensory neurons, we hypothesized that all three lines label an overlapping population of sensory neurons required for yeast feeding. To test this, we generated an *Ir76b-GAL80* transgene to suppress expression of effectors in *Ir76b*-expressing neurons. Indeed, combining *Ir76b-GAL80* with *1261-GAL4* or *Ir25a-GAL4* suppressed expression in a subset of chemosensory neurons (*Figure 1E* and *Figure 1—figure supplement 2I*). Removal of these overlapping neurons abolished the phenotype of silencing these lines, indicating that these three lines label a common population of *Ir76b-* and *Ir25a*-expressing neurons required for yeast feeding. Importantly, this phenotype was due to a sensory deficit, and not to an effect of silencing neurons in the central nervous system, since *Ir25a-GAL4* exclusively labels peripheral neurons (note absence of cell bodies in brains and VNC of *Ir25a-GAL4* animals in *Figure 1D* and *Figure 1—figure supplement 2F*). These lines therefore provide an entry point to dissect the sensory basis of yeast feeding.

Yeast is the main source of AAs for wild *Drosophila*, and recently a subset of *Ir76b*-expressing neurons in the legs has been proposed to respond to AAs through the Ir76b receptor (*Ganguly et al., 2017*). We also observed a loss of preference for AA-rich food upon silencing the lines we identified: silencing *1261-GAL4*, *Ir25a-GAL4* or *Ir76b-GAL4* abolished the preference of protein-deprived flies for a diet containing AAs over one containing sucrose (*Leitão-Gonçalves et al., 2017*) (*Figure 1—figure supplement 3A*). However, as with other yeast metabolites (*Figure 1—figure supplement 1A*), flies showed a much lower level of feeding on AAs compared to yeast (*Figure 1—figure supplement 3B*). Furthermore, mutations in *Ir76b*, which abolish AA responses (*Ganguly et al., 2017*), or in *Ir25a* had no effect on yeast feeding (*Figure 1—figure supplement 3C–E*). These data strongly suggest that AAs are not the sole stimuli mediating yeast appetite.

## Taste receptor neurons within the identified lines are required for yeast feeding

Upon further inspection, we noted that all of the lines identified above show strong expression in multiple subsets of GRNs (*Figure 1* and *Figure 1—figure supplement 2*). We therefore hypothesized that the reduction in yeast feeding we identified would depend on silencing of taste receptor neurons common to these lines. To test this, we took advantage of the gustatory-specific enhancer of *Poxn* to generate a GRN-specific GAL80 line (*Poxn-GAL80*). This *Poxn-GAL80* line has the unique property that, unlike the endogenous *Poxn* transcript or *Poxn-GAL4* (*Boll and Noll, 2002*), it is expressed broadly in the gustatory system, including taste peg GRNs. Therefore, in combination with the identified drivers, this transgene blocks the expression of effector genes in GRNs innervating taste sensilla, taste pegs, and almost all pharyngeal GRNs, while leaving expression in ORNs largely unaffected (*Figure 2A*, *Figure 2—figure supplement 1A and B*, and *Supplementary file 1*). Relieving the silencing of GRNs in any of the *GAL4* lines rescued flies' preference for yeast, suggesting that these lines label GRNs that are necessary for yeast feeding (*Figure 2A* and *Figure 2—figure supplement 1A and B*). To directly test the possible involvement of olfactory receptor neurons

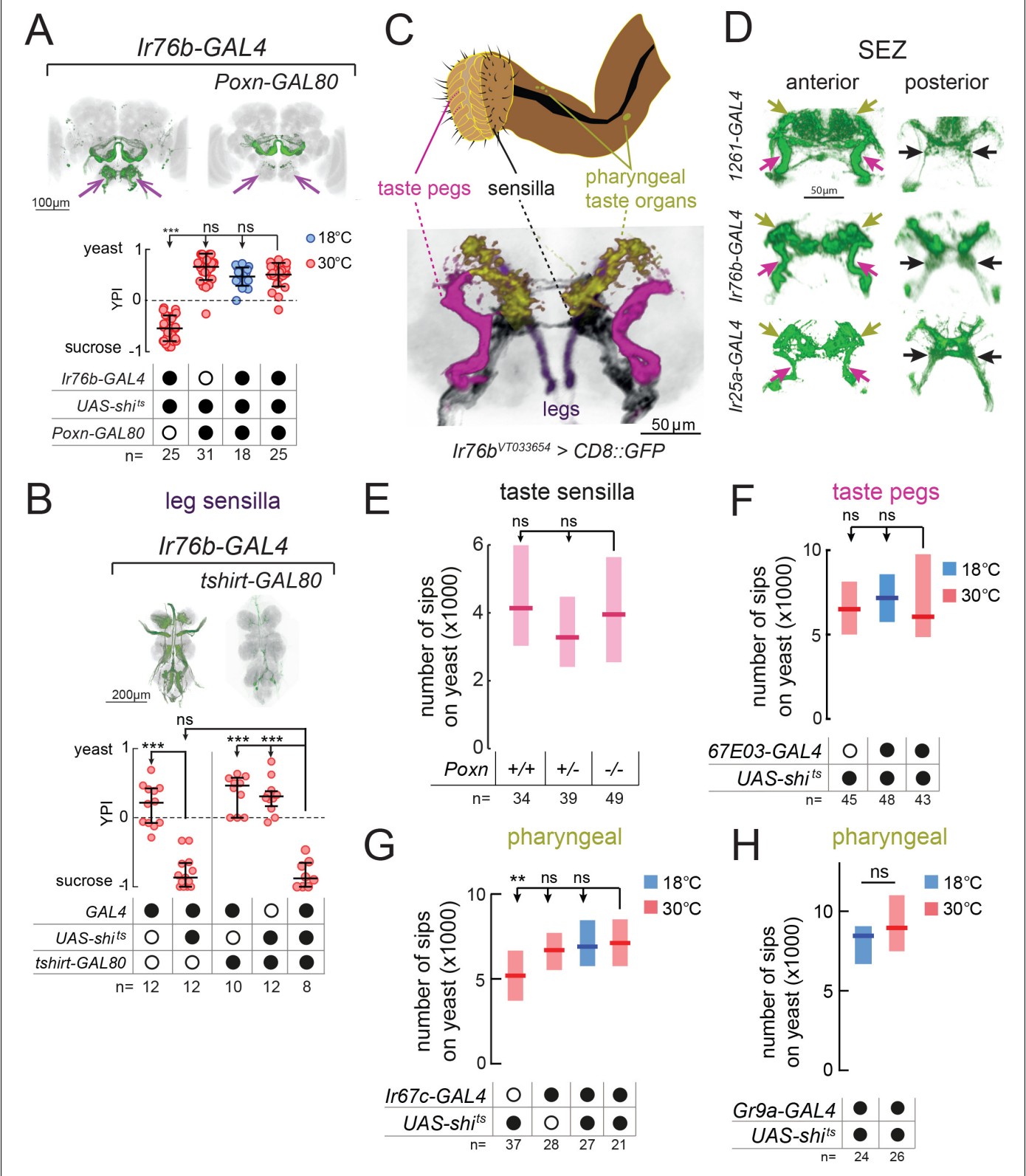

**Figure 2.** Proboscis gustatory receptor neurons are required for yeast intake. (A) YPI of yeast-deprived female flies in which *Ir76b-GAL4* neurons were all silenced, or with *Poxn-GAL80*, and corresponding controls. Expression pattern of experimental flies in the brain as visualized using *UAS-CD8::GFP* in green, with nc82 synaptic staining in gray. Arrows indicate presence/loss of GFP expression in the SEZ. (B) YPI of yeast-deprived female flies in which *Ir76b-GAL4* neurons were all silenced, or with *tshirt-GAL80*, and corresponding controls. Expression pattern of experimental flies in the VNC as
*Figure 2 continued on next page*

*Figure 2 continued*

visualized using *UAS-CD8::GFP* in green, with nc82 synaptic staining in gray. (**C**) Upper: Schematic of the proboscis, showing sensilla, taste pegs and pharyngeal taste organs. Lower: Schematic view of the SEZ of flies expressing CD8::GFP from a fragment of the *Ir76b* enhancer (*VT033654-GAL4*) with GRNs colored by their peripheral innervation and gray representing the nc82 synaptic staining. Pink, taste pegs; black, sensilla; yellow, pharyngeal taste organs; purple, legs. (**D**) Expression of *1261-*, *Ir76b-* and *Ir25a-GAL4* in the anterior (left) and posterior (right) SEZ, with arrows showing projections from pharyngeal (yellow), taste peg (pink) and sensillar (black) GRNs. (**E**) Number of sips from yeast by flies with 0/1/2 copies of the *Poxn*$^{AM22-B5}$ mutation. The homozygous mutant also contains a rescue construct to rescue all defects except taste sensilla (see Materials and methods). (**F–H**) Number of sips from 10% yeast by yeast-deprived females in which *67E03-* (**F**), *Ir67c-* (**G**) or *Gr9a-GAL4* (**H**) neurons were acutely silenced, and corresponding controls. *67E03-GAL4* labels taste peg GRNs, *Ir67c-GAL4* labels a subset of GRNs in the LSO, and *Gr9a-GAL4* labels a subset of GRNs in the VCSO. In (**A**) and (**B**), circles represent yeast preference in single assays, with line representing the median and whiskers the interquartile range. In (**E–H**), boxes represent median with upper/lower quartiles. ***p<0.001, **p<0.01, ns p≥0.05. Groups compared by Kruskal-Wallis test, followed by Dunn's multiple comparison test.

DOI: https://doi.org/10.7554/eLife.31625.007

The following figure supplements are available for figure 2:

**Figure supplement 1.** Olfactory receptor neurons labeled by the identified lines do not explain the yeast feeding phenotype.

DOI: https://doi.org/10.7554/eLife.31625.008

**Figure supplement 2.** Silencing subsets of GRNs within the identified lines does not affect yeast feeding.

DOI: https://doi.org/10.7554/eLife.31625.009

**Figure supplement 3.** Silencing subsets of pharyngeal GRNs does not affect yeast feeding.

DOI: https://doi.org/10.7554/eLife.31625.010

(ORNs) labeled by these lines, we performed multiple loss-of-function manipulations of ORNs. In accordance with the above results, none of the manipulations of ORNs, including *atonal* mutants which lack all *Ir*-expressing ORNs, had any effect on yeast preference (*Figure 2—figure supplement 1C–G*). This suggests that blockade of ORNs labeled in these lines does not account for the yeast feeding phenotype of silencing the *GAL4* lines identified above, and indicates that this phenotype is due to a loss of gustatory input.

## Proboscis gustatory receptor neurons in distinct locations act in parallel to support total yeast feeding

All of the lines we identified label GRNs in both the legs and the proboscis. To test the role of leg GRNs in the observed phenotypes, we used *tshirt-GAL80* to remove tarsal and wing GRNs from the expression pattern of *Ir76b-GAL4* (*Clyne and Miesenböck, 2008*) (*Figure 2B*). This did not suppress the phenotype of silencing this line, indicating that the GRNs required for yeast feeding reside in the proboscis.

Within the proboscis, GRNs are present in 3 types of taste organ: in taste sensilla on the external surface of the labellum; in taste pegs on the inner surface; and in pharyngeal taste organs that contact food after ingestion (*Stocker, 1994*) (*Figure 2C*). GRNs innervating these distinct structures send axonal projections to distinct regions of the SEZ (*Koh et al., 2014*; *Kwon et al., 2014*; *Miyazaki and Ito, 2010*; *Wang et al., 2004*). Within all of the lines we identified, we observed projections in the PMS4 region, which receives input largely from phagostimulatory sensillar GRNs (black arrows); the AMS1 region, which receives input largely from taste peg GRNs (pink arrows); and in the dorso-anterior SEZ, which receives pharyngeal GRN input (yellow arrows, *Figure 2D*).

We next aimed to separate the role of these distinct taste structures by manipulating each in turn. To remove taste sensilla function, we used a mutation in *Poxn* (*Boll and Noll, 2002*). We found that flies lacking taste sensilla fed on yeast to the same extent as controls, indicating that sensillar GRNs alone are dispensable for yeast feeding (*Figure 2E*). Furthermore, silencing of neurons expressing *Poxn-GAL4* (*Boll and Noll, 2002*), which labels GRNs innervating sensilla and the LSO, had no effect on total yeast feeding, but did strongly reduce feeding on sucrose (*Figure 2—figure supplement 2A–C*).

In order to gain access to taste peg GRNs, we visually inspected a database of *GAL4* lines (*Jenett et al., 2012*) and identified two lines, *67E03-* and *57F03-GAL4*, which show expression in taste peg GRNs in addition to other central neurons (*Yapici et al., 2016*) (*Figure 2—figure supplement 2D and E*). We used these lines to silence taste peg GRNs, but, as with our manipulation of sensillar GRNs, observed no effect on yeast feeding (*Figure 2F* and *Figure 2—figure supplement*

*2F*). These data are in agreement with the lack of yeast feeding phenotype observed upon silencing of either *E409-GAL4* or *Gr64e-GAL4*, which have previously been characterized to label taste peg gustatory neurons (*Fischler et al., 2007*; *Wisotsky et al., 2011*) (*Figure 1—figure supplement 1B and C*). Likewise, combining *67E03-GAL4* with *Gr5a-GAL4*, which labels a subset of sensillar GRNs (*Marella et al., 2006*; *Wang et al., 2004*), was not sufficient to decrease yeast feeding (*Figure 2— figure supplement 2G*).

Finally, to manipulate pharyngeal GRNs, we conducted a series of experiments in which we blocked output from subsets of *Gr*- and *Ir*-expressing GRNs previously shown to innervate the different pharyngeal taste organs (*Joseph et al., 2017*; *Koh et al., 2014*; *Kwon et al., 2014*) (*Figure 2G and H*, and *Figure 2—figure supplement 3*). None of these lines showed a reduction in yeast feeding compared to all controls, suggesting that the tested pharyngeal GRNs are not essential to support total yeast feeding.

Taken together, these data indicate that the GRNs in the proboscis common to the three identified lines play an important role in yeast feeding, and support the idea that the identified GRNs innervating sensilla, taste pegs and pharyngeal organs act redundantly in mediating yeast feeding, such that if one set is compromised, the others still suffice to support yeast feeding.

## Sensillar and taste peg GRNs mediate distinct behavioral programs contributing to yeast feeding

Like mammals, flies feed on solid food in discrete bouts, known as *feeding bursts*, each of which is composed of a series of *sips* (*Itskov et al., 2014*) (*Figure 3A*). The data presented above suggest that distinct subsets of *Ir76b* GRNs act redundantly to support total yeast feeding. This finding is reminiscent of the organization of sweet taste in *Drosophila*, whereby sweet-sensing GRNs in distinct anatomical locations are thought to act redundantly to determine total sugar intake (*LeDue et al., 2015*). In this case, however, these distinct subsets perform different functions in sugar feeding: while sensillar GRNs mediate proboscis extension in response to sugars (*LeDue et al., 2015*; *Wang et al., 2004*), pharyngeal neurons are important for sustaining sugar feeding after this point (*LeDue et al., 2015*). We therefore hypothesized that distinct subsets of yeast GRNs might similarly be involved in stimulating distinct behavioral programs that contribute to food consumption. We first focused on taste pegs, since these structures have been hypothesized by Vincent Dethier to sustain feeding in blowflies (*Dethier, 1976*). To identify behavioral programs mediated by taste pegs, we investigated how silencing these GRNs affected the microstructure of yeast feeding behavior, using the ability of the flyPAD to dissect the microstructure of feeding (*Figure 3B*). We found that silencing taste peg GRNs, using *67E03-* or *57F03-GAL4*, resulted in a reduction in the mean number of sips in each feeding burst (*Figure 3C and D*) without altering total yeast intake (*Figure 2F* and *Figure 2—figure supplement 2F*). These results suggest that taste peg GRNs may be important for sustaining yeast feeding bursts after they have been initiated. The number of sips per sucrose feeding burst, in contrast, was not affected by silencing taste peg GRNs (*Figure 3—figure supplement 1A and B*), indicating that this effect is specific to yeast taste.

To directly test the hypothesis that taste peg neurons play a specific role in sustaining yeast feeding, we aimed to silence the activity of these neurons exclusively following the initiation of feeding bursts. To do so, we employed a closed-loop optogenetic approach to silence these neurons in a temporally precise manner only after feeding has been initiated (*Figure 3E*). Flies were placed in a flyPAD arena containing two yeast patches, and interactions with these food patches were analyzed in real time. Each flyPAD arena was equipped with a green LED, and interaction with one of the yeast patches was programmed to trigger the activation of this LED after a specific delay. This experimental design should result in the silencing of neurons expressing the anion channelrhodopsin GtACR1 (*Govorunova et al., 2015*; *Mohammad et al., 2017*) after the initiation of a feeding burst. Contact with the other yeast patch was not linked to LED activation, providing an internal control for each fly. We first tested our hypothesis that yeast sensory neuron activity after feeding burst initiation is required to sustain feeding. To do so, we used this optogenetic approach to silence the activity of neurons labeled by *1261-* or *Ir76b-GAL4* after burst initiation. Indeed, such temporally precise inactivation of these neurons led to a reduction in the number of sips per yeast feeding burst, which was absent in genetic controls, confirming that these lines label neurons required for sustaining yeast feeding (*Figure 3—figure supplement 1C and D*).

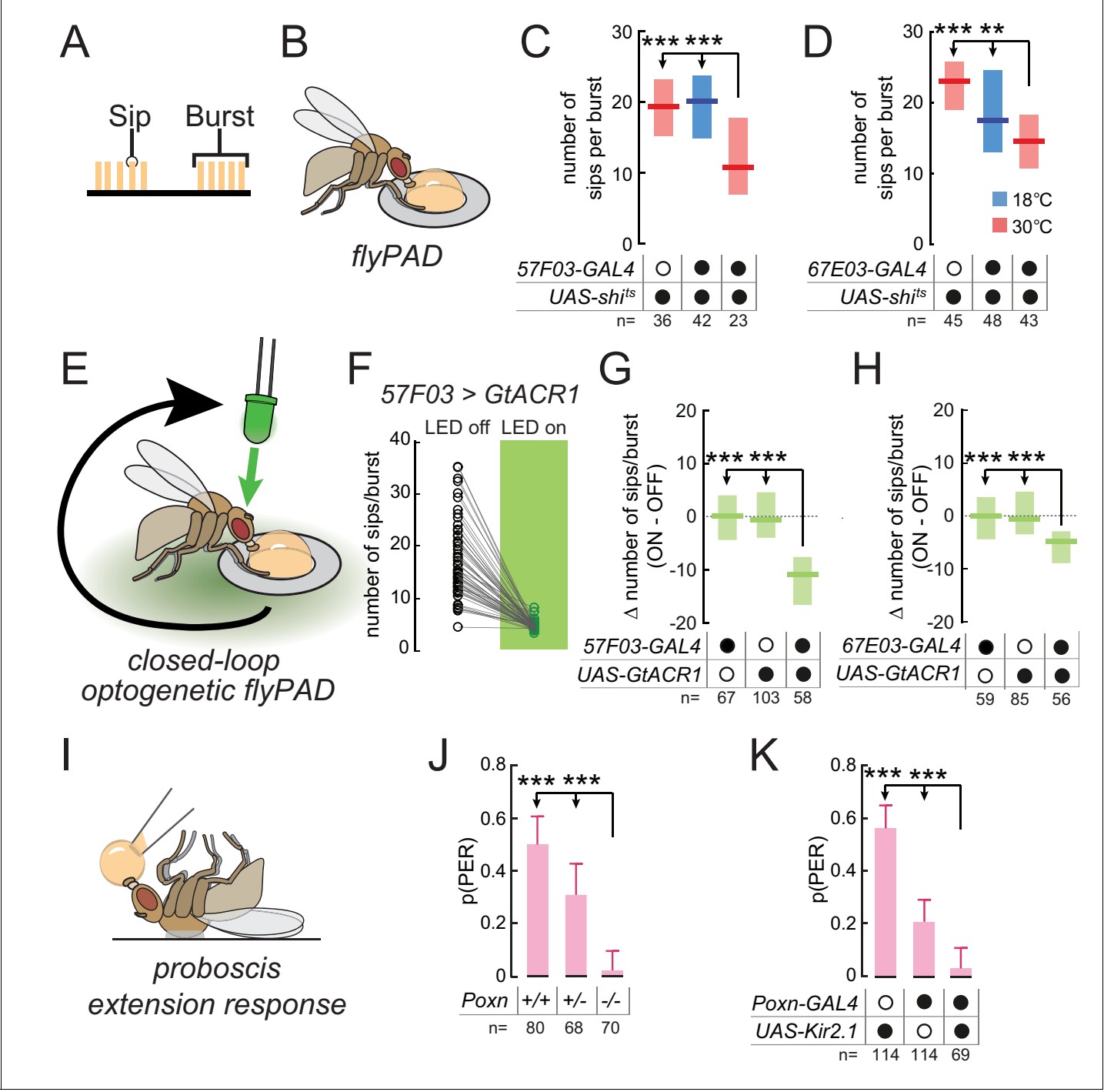

**Figure 3.** Taste peg and sensillar GRNs sustain and initiate yeast feeding, respectively. (A) Schematic of the microstructure of feeding behavior. Sips are grouped into feeding bursts, which can be detected using the flyPAD. (B) Representation of the flyPAD assay. (C) and (D) Number of sips per burst from 10% yeast by yeast-deprived females expressing *shibire*[ts] under the control of the indicated *GAL4* lines, which label taste peg GRNs, and corresponding controls. (E) Schematic of closed-loop optogenetic silencing of GRNs. Interaction with one of the two food patches leads to activation of a green LED after a delay, resulting in silencing of neurons expressing *GtACR1* after feeding initiation. (F) Number of sips per feeding burst from the unstimulated (LED off) and the light-stimulated (LED on) yeast patches by flies expressing *GtACR1* under the control of *57F03-GAL4*. (G) and (H) Difference in number of sips per feeding burst on the stimulated (ON) and unstimulated (OFF) yeast patches for flies expressing *GtACR1* under the control of the indicated *GAL4* lines, which label taste peg GRNs, and corresponding genetic controls. (I) Representation of the proboscis extension response (PER) assay. (J) Probability of PER in response to 10% yeast presented to the labellum, by flies lacking taste sensilla due to *Poxn* loss-of-function, and genetic controls. For full genotypes, see Materials and methods. (K) Probability of PER in response to 10% yeast presented to the

*Figure 3 continued on next page*

*Figure 3 continued*

labellum, by flies expressing *Kir2.1* under the control of *Poxn-GAL4* and genetic controls. \*\*\*p<0.001, \*\*p<0.01, ns p≥0.05. In (**C**), (**D**), (**G**) and (**H**), boxes represent median with upper/lower quartiles; groups compared by Kruskal-Wallis test, followed by Dunn's multiple comparison test. In (**F**), circles represent individual flies, and lines link data points from the two food patches for each single fly. In (**J**) and (**K**), bars represent fraction of flies producing PER with 95% confidence interval; groups compared by Fisher's exact test.

DOI: https://doi.org/10.7554/eLife.31625.011

The following figure supplement is available for figure 3:

**Figure supplement 1.** Taste peg GRNs specifically sustain yeast feeding bursts, and flies compensate for the loss of peg GRNs.

DOI: https://doi.org/10.7554/eLife.31625.012

We then tested whether taste peg neurons mediate this sustaining of feeding bursts by specifically silencing taste peg GRNs after feeding initiation. We found that this precisely timed silencing of taste peg GRNs, using *57F03-* or *67E03-GAL4*, resulted in a drastic reduction in the number of sips per yeast feeding burst on the stimulated compared to the unstimulated channel (*Figure 3F–H*). We further confirmed the role of taste peg GRNs in sustaining feeding bursts using *E409-GAL4*, which has been previously shown to specifically label taste peg GRNs (*Fischler et al., 2007*) (*Figure 3—figure supplement 1E*). Importantly, this reduced burst length was specific to flies expressing GtACR1, and not genetic controls. The phenotype of *57F03-GAL4* was not due to labeling of IN1 neurons (*Yapici et al., 2016*), since acute silencing using *83F01-GAL4*, which also labels IN1 but not taste peg neurons (*Yapici et al., 2016*), did not affect yeast feeding bursts (*Figure 3—figure supplement 1F*). Likewise, silencing sweet sensillar neurons in the same manner, using *Gr5a-GAL4* (*Wang et al., 2004*), did not affect the number of sips per yeast feeding burst (*Figure 3—figure supplement 1G*), indicating that this effect was specific to taste peg GRNs. These results confirm the finding that the activity of taste peg GRNs is important to sustain yeast feeding bursts after they have been initiated.

In contrast to taste peg GRNs, we hypothesized that sensillar GRNs might play a role in initiating feeding in response to yeast taste. This hypothesis was based on the anatomical location of taste sensilla, which are situated on the outer surface of the labellum and therefore come into contact with yeast early in the feeding program; and on the fact that sensillar GRNs mediate proboscis extension in response to sugars (*LeDue et al., 2015*; *Wang et al., 2004*). To test this hypothesis, we turned to the proboscis extension response (PER), an innate response of flies to stimulation of the labellum with attractive tastants and which is generally thought to underlie the initiation of feeding bursts (*Figure 3I*). We found that *Poxn* mutants, which lack taste sensilla but retain taste peg and pharyngeal GRNs (*Boll and Noll, 2002*; *LeDue et al., 2015*), showed a near-complete loss of PER to yeast when presented to the labellum (*Figure 3J*). This result extends previous work showing that PER in response to yeast presented to the tarsi is abolished in *Poxn* mutants (*Masek and Keene, 2013*). Likewise, silencing of *Poxn-GAL4*, which labels most sensillar GRNs in addition to some GRNs in the LSO (*Figure 2—figure supplement 2A*), drastically reduced the probability of PER to yeast (*Figure 3K*). These results indicate that while taste peg GRNs are important for sustaining yeast feeding bursts, sensillar GRNs mediate proboscis extension in response to yeast taste.

It is important to note that while loss of sensillar or taste peg GRN function did affect specific aspects of the feeding program, flies in which these GRN subsets were compromised were still able to attain the same total level of yeast feeding as controls (*Figure 2E and F*). This reinforces our hypothesis that feeding behavior is highly plastic: when one aspect of feeding is impaired, flies will compensate by altering their pattern of feeding to ensure sufficient uptake of critical nutrients. Indeed, we observe evidence for such compensatory changes in flies in which taste peg GRNs are silenced. These flies compensated for the reduced length of yeast feeding bursts by increasing the total number of yeast feeding bursts (*Figure 3—figure supplement 1H and I*). Given that protein is an essential resource for *Drosophila*, a strategy which relies on distributed gustatory structures linked with behavioral programs that can be flexibly organized ensures that the fly can reach the required level of yeast intake. Since both sensillar and taste peg GRNs within the identified lines seem to play important roles in yeast feeding, we predicted that both should respond to the taste of yeast.

## *Ir76b* labellar GRNs respond to yeast taste

We have shown that *Ir76b*- and *Ir25a*-expressing GRNs in the proboscis of the fly are required for yeast intake. To characterize the response of these neurons to yeast taste, we performed two-photon calcium imaging of *Ir76b* GRN axons in the SEZ of awake flies while stimulating the labellum with liquid stimuli (*Figure 4A*). To capture the response in all the projection fields of GRN axons, we imaged at multiple planes spanning the antero-posterior extent of the SEZ (*Figure 4B*). In agreement with our behavioral data indicating that both sensillar and taste peg GRNs participate in yeast feeding, we observed strong calcium responses to yeast in both the AMS1 and PMS4 regions, where axons of neurons innervating the taste pegs and taste sensilla, respectively, terminate (*Miyazaki and Ito, 2010*) (*Figure 4C and D*; *Video 1*). These yeast responses were significantly greater than responses to water or 500 mM sucrose (*Figure 4E and F*), and were sharply aligned to stimulus onset and offset (*Figure 4—figure supplement 1A and B*; *Video 1*). *Ir76b*-positive taste peg and sensillar neurons therefore preferentially respond to yeast taste, supporting the view that multiple subsets of GRNs on the labellum are involved in yeast feeding.

The response to yeast in AMS1 is in agreement with earlier data indicating that taste peg GRNs respond to carbonation, a by-product of yeast metabolism (*Fischler et al., 2007*). We confirmed that taste peg GRNs responded to carbonation; however, we found that these GRNs also responded

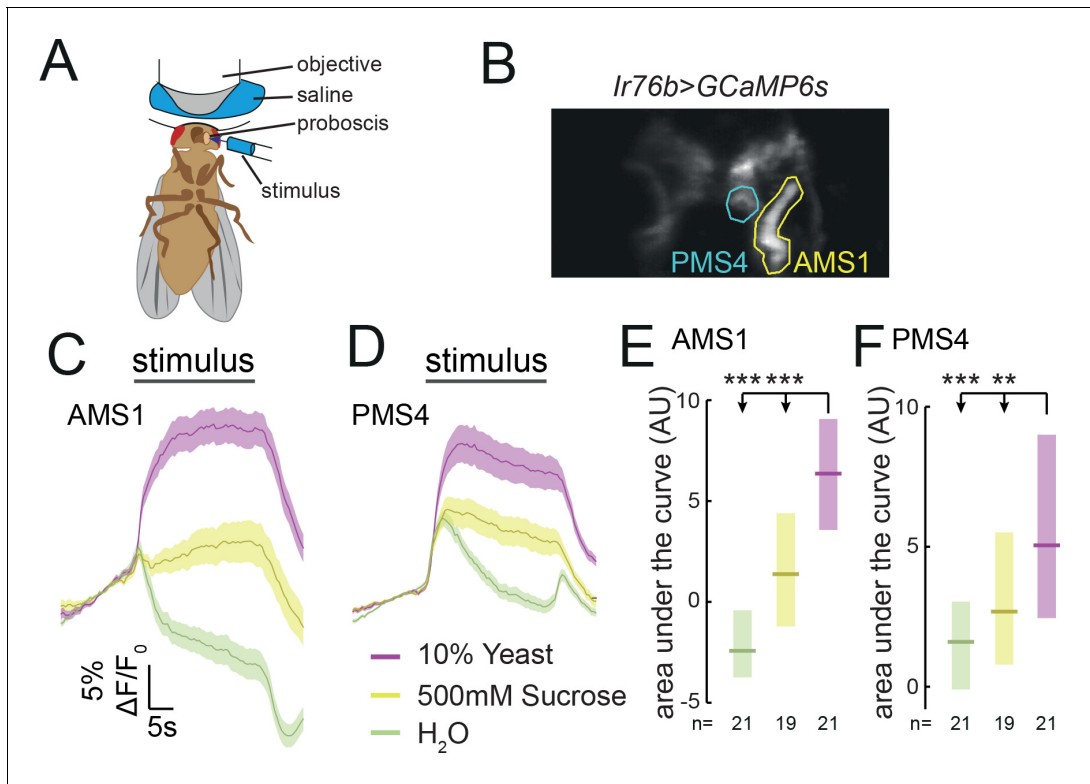

**Figure 4.** *Ir76b* GRNs respond to yeast taste. (**A**) Schematic of the imaging setup. The fly is head-fixed, with a window in the head to allow visual access to GRN axons in the SEZ using a two-photon microscope. The fly is stimulated on the labellum with liquid tastant solutions. (**B**) Representative z-projection of baseline GCaMP6s fluorescence from *Ir76b-GAL4* axons in the SEZ. Highlighted ROIs indicate the AMS1 and PMS4 regions, which are largely innervated by taste peg and sensillar GRNs, respectively. (**C–F**) Average responses measured in ROIs shown in (**B**) to 10% yeast (purple), 500 mM sucrose (yellow) or water (green) from females deprived of yeast for 10 days. Average (mean ± SEM) trace of $\Delta F/F_0$ from GCaMP6s signal in AMS1 (**C**) and PMS4 (**D**) upon taste stimulation. Black line indicates stimulus period. Responses quantified as area under the curve in AMS1 (**E**) and PMS4 (**F**) during stimulus presentation. Boxes represent median with upper/lower quartiles. **p<0.01, ***p<0.001. Groups compared by Kruskal-Wallis test, followed by Dunn's multiple comparison test.

DOI: https://doi.org/10.7554/eLife.31625.013

The following figure supplement is available for figure 4:

**Figure supplement 1.** *Ir76b* GRNs respond to yeast metabolites.
DOI: https://doi.org/10.7554/eLife.31625.014

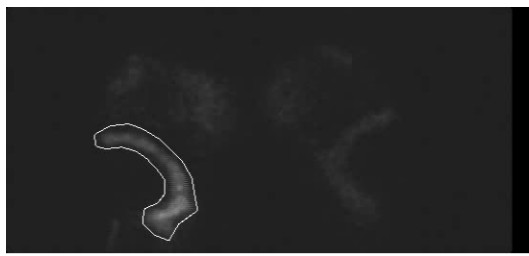

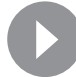

**Video 1.** *Ir76b* taste peg GRNs respond to yeast taste. Representative time series of yeast response. Upper: time series of average z-projections of GCaMP6s fluorescence from *Ir76b-GAL4* axons in the SEZ. Highlighted ROI indicates the AMS1 region in one hemisphere, which is innervated by taste peg GRNs. Lower: trace of average ΔF/F$_0$ value within the highlighted ROI. Yeast is presented to the labellum and removed at the specified time points, indicated by blue and orange lines, respectively. Movie is at 10x real time.

DOI: https://doi.org/10.7554/eLife.31625.015

to deactivated yeast, which does not produce carbonation, suggesting that taste peg GRNs can detect yeast independently from carbonation (*Figure 4—figure supplement 1C and D*). Likewise, the response in PMS4 is consistent with reported activation of sensillar GRNs by glycerol (*LeDue et al., 2015*; *Wisotsky et al., 2011*), though the response to glycerol concentrations in the range found in yeast are significantly smaller than the response to yeast (*Scanes et al., 2017*) (*Figure 4—figure supplement 1E and F*). Furthermore, we observed an increase in calcium in response to putrescine in PMS4, but not in AMS1 (*Hussain et al., 2016b*) (*Figure 4—figure supplement C and D*). AA solutions, likewise, evoked a much smaller responses than yeast in both AMS1 and PMS4 (*Figure 4—figure supplement G and H*). This is consistent with the data presented above suggesting that AA taste alone is not a major stimulus driving yeast feeding (*Figure 1—figure supplement 3B*). Together, these data indicate that yeast metabolites activate distinct subsets of *Ir76b*-positive neurons, which is consistent with our behavioral data indicating that distinct subsets of GRNs contribute to yeast feeding (*Figures 2* and *3*).

## The response of yeast GRNs is modulated by internal amino acid state

In order to direct feeding decisions towards achieving nutrient homeostasis, animals must integrate information about their current needs with sensory information from foods available in the environment. Mated female flies, when deprived from dietary yeast, respond with a homeostatic increase in yeast feeding (*Ribeiro and Dickson, 2010*) (*Figure 5A* and *Figure 1—figure supplement 2A*). This appetite is yeast-specific, since sucrose feeding is not increased by yeast deprivation (*Figure 5C*). Flies are therefore able to adjust their food intake in a nutrient-specific fashion. We speculated that internal nutrient state changes, induced by deprivation from yeast, may specifically affect the response to yeast taste at the level of sensory neurons. To test this, we imaged the responses of *Ir76b* GRNs to yeast taste in flies fed on sucrose alone for varying durations. We found that 3 days of yeast deprivation resulted in a small but non-significant increase in the response of both taste peg and sensillar GRNs to yeast, and that this response was significantly enhanced after 10 days of yeast deprivation (*Figure 5B* and *Figure 5—figure supplement 1A*). The responses of *Ir76b* GRNs to the water solvent, in contrast, were not increased by yeast deprivation (*Figure 5—figure supplement 1B and C*).

If this sensory modulation is part of a nutrient-specific appetite, it should be specific to GRNs that support feeding on yeast, and not other taste stimuli. Indeed, we found that the response of sugar-sensing GRNs (*Gr5a-GAL4*) to sucrose was not enhanced by yeast deprivation (*Figure 5D*). Surprisingly, deprivation from dietary yeast led to a decrease in feeding on sucrose (*Figure 5C*), which was paralleled by a decrease in the response of *Gr5a* GRNs to low sucrose (20 mM, the same concentration used in behavioral experiments, *Figure 5—figure supplement 1D*). This reinforces the notion that yeast deprivation is distinct from starvation, which has been shown to increase the sugar sensitivity of *Gr5a* neurons (*Inagaki et al., 2012*). Thus, deprivation from a specific food, yeast,

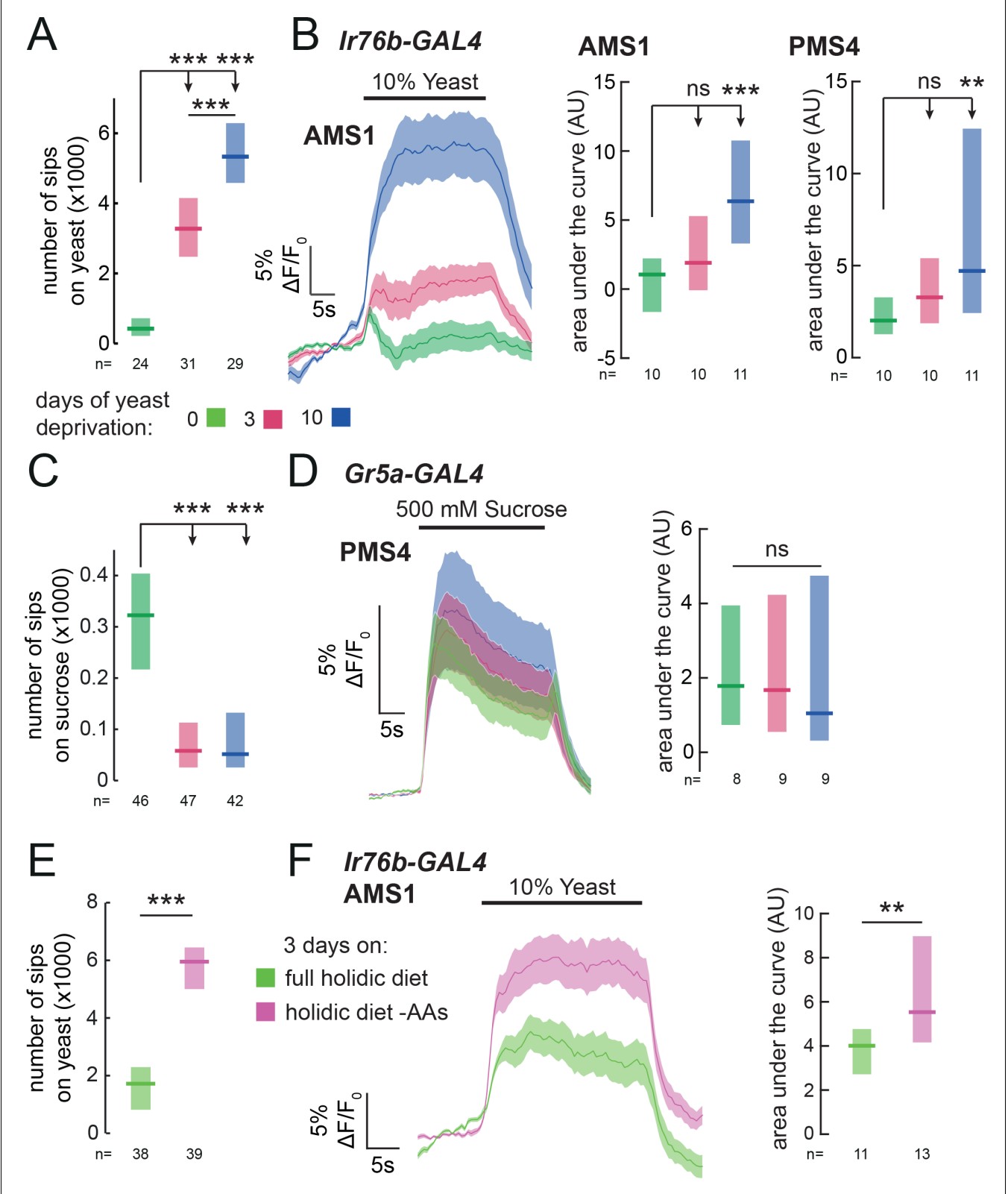

**Figure 5.** Yeast or amino acid deprivation modulates yeast feeding and enhances the response of yeast GRNs. (**A**) and (**C**) Number of sips from 10% yeast (**A**) and 20 mM sucrose (**C**) by female flies of the genotype *Ir76b-GAL4,UAS-GCaMP6s*. Flies were deprived from yeast for 0, 3 or 10 days. (**B**) Left: Average (mean ± SEM) trace of $\Delta F/F_0$ from GCaMP6s signal in *Ir76b-GAL4* AMS1 region upon stimulation of the labellum with 10% yeast, from flies deprived of yeast for 0, 3 or 10 days. Quantification of responses of AMS1 (center) and PMS4 (right) regions as area under the curve during stimulus

*Figure 5 continued on next page*

*Figure 5 continued*

presentation. (D) Left: Average (mean ± SEM) trace of $\Delta F/F_0$ measured from GCaMP6s in *Gr5a-GAL4* taste sensillar projections upon stimulation of the labellum with 500 mM sucrose, from flies deprived of yeast for 0, 3 or 10 days. Right: Quantification of responses as area under the curve during stimulus presentation. (E) Number of sips from 10% yeast by females fed on a holidic diet with or without amino acids for 3 days prior to assay. (F) Left: Average (mean ± SEM) trace of $\Delta F/F_0$ from GCaMP6s signal in *Ir76b-GAL4* AMS1 region upon stimulation of the labellum with 10% yeast, from flies treated as in (E). Right: Quantification of responses as area under the curve during stimulus presentation. (A), (C) and (E) Feeding behavior of mated female flies expressing GCaMP6s under the control of *Ir76b-GAL4* was measured on the flyPAD. Boxes represent median with upper/lower quartiles. ***p<0.001, **p<0.01, *p<0.05, ns p≥0.05. (A-D) Groups compared by Kruskal-Wallis test, followed by Dunn's multiple comparison test. (E-F) Groups compared by Wilcoxon rank-sum test.

DOI: https://doi.org/10.7554/eLife.31625.016

The following figure supplement is available for figure 5:

**Figure supplement 1.** Effects of yeast deprivation on taste responses.
DOI: https://doi.org/10.7554/eLife.31625.017

specifically enhances the sensitivity of primary sensory neurons detecting that nutrient source, providing a potential basis for homeostatic changes in nutrient choice.

Of all the nutrients present in yeast, AAs are the most potent modulators of reproduction and lifespan, critical life-history traits of the animal (*Piper et al., 2014*). It is therefore unsurprising that internal AA state is the main nutritional factor regulating yeast appetite (*Leitão-Gonçalves et al., 2017*). We hypothesized that the increase in sensory gain to yeast would be driven by the internal AA state of the fly, such that GRNs would have increased gain when the fly is low on AAs. Alternatively, exposure to dietary yeast could desensitize yeast GRNs, so that a diet devoid of yeast would increase their sensitivity independently of the flies' nutritional state. To disentangle the effects of AA state from sensory experience, we turned to a chemically defined diet, which allowed us to manipulate specific components of the diet independently, in the absence of exposure to yeast (*Corrales-Carvajal et al., 2016*; *Piper et al., 2014, 2017*). As expected, removal of AAs from the flies' diet elicited a specific appetite for yeast (*Figure 5E*). We then imaged the response of yeast-sensing GRNs in flies deprived specifically from AAs, and found that the gain of these GRNs is enhanced by AA deprivation (*Figure 5F*). These data indicate that the internal AA state of the fly modulates the gain of yeast-sensing GRNs, potentially allowing the fly to homeostatically compensate for the lack of AAs.

## Reproductive state acts downstream of sensory neurons to modulate yeast feeding

Animals have to constantly integrate information from multiple internal states to produce adaptive behaviors. Accordingly, yeast appetite is regulated not only by AA state, but also by the reproductive state of the fly (*Corrales-Carvajal et al., 2016*; *Ribeiro and Dickson, 2010*; *Vargas et al., 2010*; *Walker et al., 2015*). Mating disinhibits yeast appetite through a dedicated circuit, such that mated females consume significantly more yeast than virgins (*Walker et al., 2015*) (*Figure 6A*). Our finding that yeast feeding is mediated by specific GRNs strongly suggests that, as with salt taste (*Walker et al., 2015*), mating changes the behavioral response to yeast taste. We therefore hypothesized that the mating state of the animal could be integrated with nutrient state to drive yeast appetite in three distinct manners: by directly modulating the response of yeast-sensing GRNs; by modulating the sensitivity of the nervous system to protein deprivation; or by modulating downstream processing independently of AA state (*Figure 6B*). Having a cellular readout for the effect of these internal states on yeast taste responses allows us to distinguish these hypotheses. We imaged the response of yeast GRNs in virgin and mated females in different yeast deprivation regimes. While yeast deprivation had a strong impact on taste peg and sensillar GRN responses to yeast, there was no significant effect of mating on these responses (*Figure 6C and D*). This is in contrast to the synergistic interaction between deprivation and mating states seen in the flies' feeding behavior, and suggests that reproductive state may act on downstream gustatory processing to influence flies' feeding on yeast. In support of this hypothesis, we found that knockdown of *Sex Peptide Receptor* in *Ir76b*-expressing neurons had no effect on females' postmating yeast appetite (*Figure 6—figure supplement 1A*), reinforcing the concept that Sex Peptide acts through the SPSN-SAG pathway, and not directly on yeast-sensing neurons, to stimulate yeast appetite in anticipation of the demands

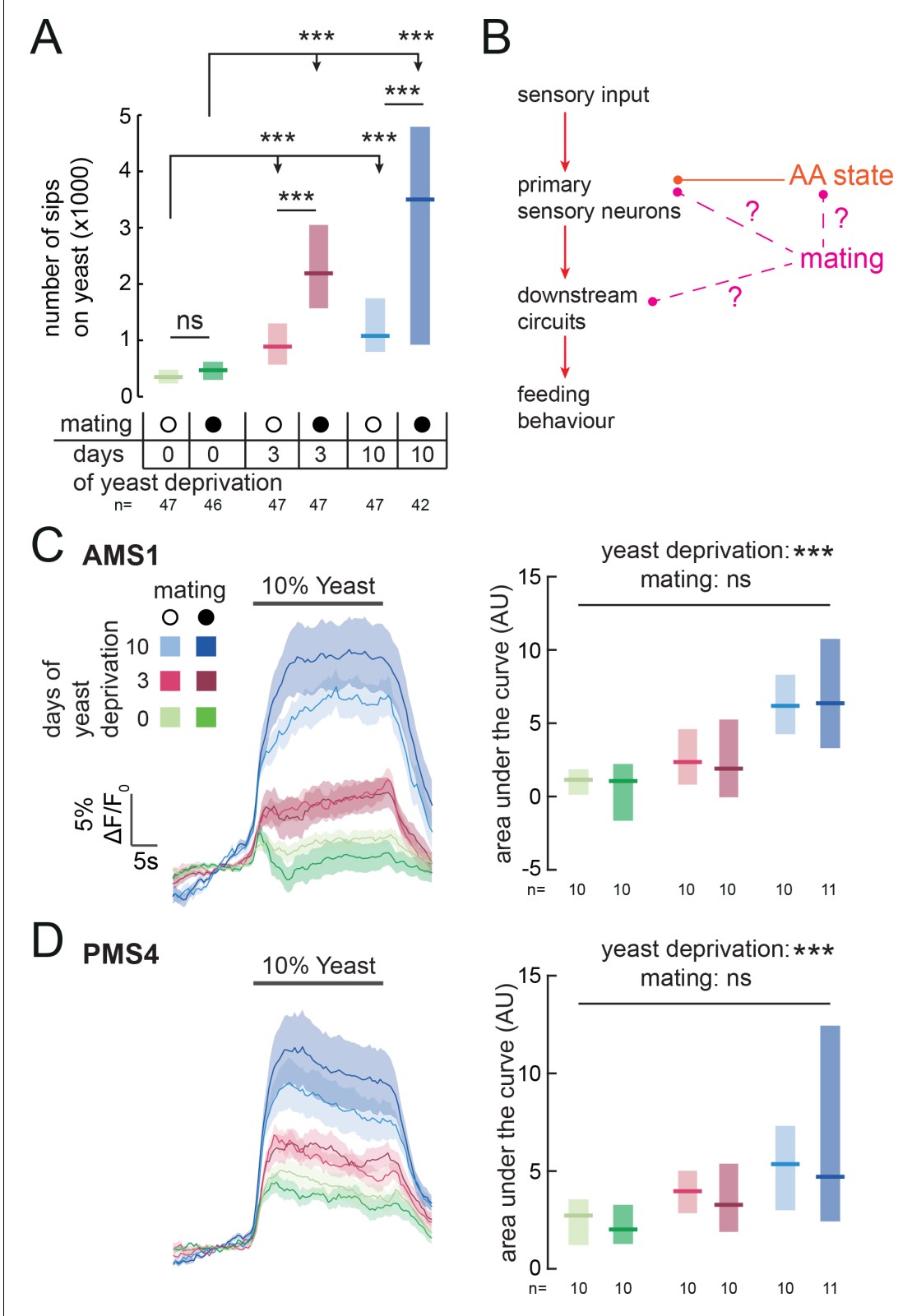

**Figure 6.** Mating state affects yeast feeding behavior but does not modulate GRN responses to yeast. (**A**) Number of sips from 10% yeast by female flies of the genotype *Ir76b-GAL4,UAS-GCaMP6s*. Flies were virgin or mated, and yeast-deprived for 0, 3 or 10 days. (**B**) Schematic of how mating could act to modulate yeast feeding behavior. Mating could regulate the sensitivity of the nervous system to AA deprivation; it could act on yeast GRNs independently from protein state; or it could act on downstream yeast taste processing circuits. (**C**) and (**D**) Calcium responses of AMS1 (**C**) and PMS4

*Figure 6 continued on next page*

*Figure 6 continued*

(D) regions of *Ir76b-GAL4* upon stimulation of the labellum with 10% yeast, from virgin and mated females deprived of yeast for 0, 3 or 10 days. Left: Average (mean ± SEM) trace of $\Delta F/F_0$. Right: Quantification of responses as area under the curve during stimulus presentation. Boxes represent median with upper/lower quartiles. Significance was tested using two-way ANOVA, with deprivation and mating states as the independent variables. In (A), this was followed by multiple comparisons with Bonferroni correction. \*\*\*$p < 0.001$, ns $p \geq 0.05$.

DOI: https://doi.org/10.7554/eLife.31625.018

The following figure supplement is available for figure 6:

**Figure supplement 1.** Knockdown of *Sex peptide receptor* in *Ir76b* neurons does not affect yeast appetite.

DOI: https://doi.org/10.7554/eLife.31625.019

---

of egg production (*Ribeiro and Dickson, 2010*; *Walker et al., 2015*, *2017*; *Carvalho-Santos and Ribeiro, 2017*). Furthermore, by pooling together virgin and mated female imaging data, we found that the response of yeast-sensing GRNs was significantly increased following just 3 days of protein deprivation (AMS1: $p = 0.004$; PMS4: $p = 0.005$, AUC). Taken together, these data show that the nervous system has independent mechanisms for detecting nutritional and reproductive states, and that these two states are integrated independently at different levels of the yeast feeding circuit.

## Discussion

The utility of environmental resources to animals is dependent on their current internal states. As such, animals must adapt their decision-making depending on these states, choosing resources that fulfill current organismal requirements while minimizing the negative impact of mismatches. In the context of feeding, this means not only regulating caloric intake, but also balancing the intake of different nutrients (*Simpson and Raubenheimer, 2012*). In this study, we identify resource-specific modulation of primary sensory neurons as a potential mechanism underlying such state-dependent tuning of value-based decisions. Specifically, we show that the amino acid (AA) state of the fly modulates the gain of gustatory sensory neurons that mediate the ingestion of yeast, flies' main source of dietary protein, such that the response of these neurons to yeast is increased when the fly is lacking AAs (*Figure 7*). The response of sensory neurons detecting sugars, in contrast, is not increased by protein deprivation, indicating that this modulation is resource-specific. Rather, the response of sugar-sensing neurons is increased by complete starvation (*Inagaki et al., 2012*). Thus, the gain of these two classes of attractive taste-coding neurons is separately regulated to drive nutrient-specific appetites.

Yeast appetite is driven by the lack of dietary essential AAs (*Corrales-Carvajal et al., 2016*; *Leitão-Gonçalves et al., 2017*; *Piper et al., 2014*), and AA state modulates GRN responses to yeast. However, the mechanism through which AA state regulates GRN gain is not known and will be an important avenue for future research. AA state could act at two levels in yeast GRNs: at presynaptic terminals within the SEZ, or at the peripheral level, to adapt GRN responses to the internal state. Recent studies in *Drosophila* have indicated that complete starvation state acts through distinct neuromodulatory mechanisms on the presynaptic terminals of sensory neurons in the brain to increase attraction to food odors (*Root et al., 2011*) and sugars (*Inagaki et al., 2012*; *Marella et al., 2012*) and to decrease sensitivity to bitter deterrents (*Inagaki et al., 2014*; *LeDue et al., 2016*). Intriguingly, acute blockade of synaptic release from these neuromodulatory systems did not suppress yeast appetite induced by yeast deprivation (*Supplementary file 1*), suggesting that a distinct mechanism may regulate sensory gain according to AA state. Alternatively, this modulation could also occur at the level of peripheral sensory responses, as seen for AA responses in the fish olfactory and locust gustatory systems (*Nikonov et al., 2017*; *Simpson and Simpson, 1992*). In locusts, this peripheral modulation occurs through desensitization by AAs in the hemolymph. Regardless of the mechanism, our results demonstrate nutrient-specific modulation of the gain of select sensory neuron responses as an elegant way to increase the salience of the specific resources that are important to maintain homeostasis in the current state, and thus to optimize lifespan and reproduction.

AA state is likely to act at multiple levels in the nervous system, in addition to its effect on primary sensory neurons, to drive a robust yeast appetite - similarly to the multilevel integration seen in other systems (*Ohyama et al., 2015*). Recently, two central systems have been proposed to modulate protein appetite in *Drosophila*: the dopaminergic WED-FB-LAL circuit, which promotes yeast intake in

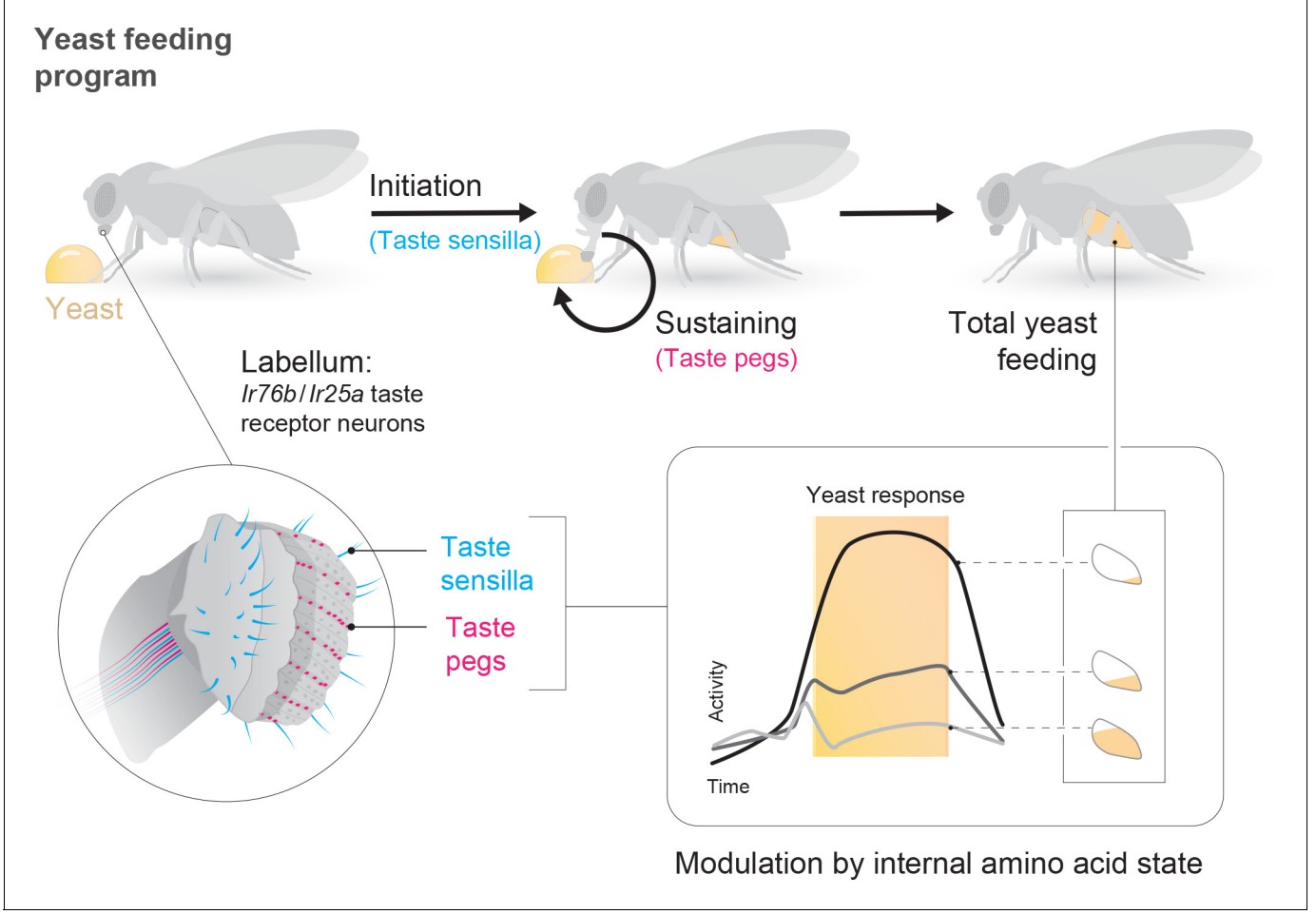

**Figure 7.** State-dependent modulation of yeast taste neurons regulates specific yeast feeding behavior programs to support protein homeostasis. Schematic depicting the regulation of yeast feeding by specific taste receptor neurons. GRNs in the proboscis that express *Ir76b* and *Ir25a* mediate yeast feeding. GRNs innervating taste sensilla on the labellum regulate proboscis extension in response to yeast, and subsequent extension and retraction cycles are regulated by taste peg GRNs. The response of each of these GRN subsets is modulated according to the fly's internal amino acid state: after 3 or 10 days of deprivation from yeast, the response of these neurons is increased. This modulation provides a potential neuronal basis to increase both the number and duration of yeast feeding bursts when the fly is lacking amino acids, and thereby to maintain protein homeostasis.
DOI: https://doi.org/10.7554/eLife.31625.020

protein-deprived flies (*Liu et al., 2017*), and the fat body-derived hormone FIT, which acts in the brain to suppress intake of protein-rich food in the sated state (*Sun et al., 2017*). Identifying whether these systems modulate sensory processing, or affect other aspects of the complex behavioral changes that guide protein homeostasis (*Corrales-Carvajal et al., 2016*), will be important for a mechanistic understanding of the circuit basis of protein appetite.

The dynamics of value-based decisions are shaped by multiple ongoing internal states, which must interact in the nervous system. How different states interact at the circuit level to shape behavior is poorly understood. Feeding on yeast is known to be synergistically modulated by both AA state and reproductive state (*Corrales-Carvajal et al., 2016*; *Ribeiro and Dickson, 2010*; *Vargas et al., 2010*; *Walker et al., 2015*). Here, we demonstrate that these two states are separately detected by the nervous system and influence taste processing at different levels of the sensory circuit. While AA state strongly modulates the gain of neurons mediating yeast feeding, reproductive state has no effect on these responses. Although virgin and mated females show very different behavioral responses to protein deprivation, our results suggest that they are subject to the same lack of AAs, and that their sensory neurons are modulated in a similar way by protein deprivation. Mating is therefore likely to gate downstream gustatory processing to enhance yeast

appetite in anticipation of the nutritional demands of egg production (*Walker et al., 2017*). This result stands in contrast to the previously-described modulation of *Ir76b⁺* GRN responses to poly-amines in the AMS1 region that occurs transiently following mating (*Hussain et al., 2016a*). This modulation is dependent on SPR expression in GRNs; however, in the context of yeast feeding, we show that SPR knockdown in *Ir76b+* neurons has no effect. Rather, mating state is detected by Sex Peptide sensory neurons in the reproductive tract, and conveyed to the brain by SAG neurons to modulate yeast and salt appetite, with the postmating yeast appetite also requiring the action of octopamine (*Corrales-Carvajal et al., 2016*; *Rezával et al., 2014*; *Walker et al., 2015*). How the activity of SAG neurons modulates downstream gustatory processing, however, remains to be discovered.

Yeast is a critical component of the diet of *Drosophila melanogaster*, providing most dietary proteins in addition to many other non-caloric nutrient requirements (*Baumberger, 1919*; *Starmer and Lachance, 2011*). Here, we identify a population of gustatory receptor neurons that is necessary for yeast feeding and responds to yeast taste (*Figure 7*). These *Ir76b-* and *Ir25a*-expressing neurons are distributed across the proboscis. The strong yeast feeding phenotype, the anatomical overlap of the lines we identified, as well as the strong and selective response of these neurons to yeast, and our finding that they are regulated by yeast deprivation, indicates that the identified neurons play a key role in yeast intake.

While different yeast GRNs seem to act redundantly to support overall yeast feeding, we demonstrate that distinct subsets of sensory neurons play different roles in the yeast feeding program: while taste sensilla are essential for proboscis extension in response to yeast, taste peg neurons play a key role in sustaining yeast feeding bursts (*Figure 7*). We show that silencing taste peg neurons specifically after feeding initiation leads to premature termination of yeast feeding bursts. These data allow us to assign a novel function to *Drosophila* taste peg GRNs: that of regulating the length of yeast feeding bursts. It is interesting to note that the anatomical location of the taste pegs, on the inner surface of the labellum, places them in an ideal position to survey the quality of the food which is being ingested upon initiation of food intake. A similar function in sustaining the intake of sweet solutions has been demonstrated for pharyngeal taste neurons, which are thought to survey the intake of sugars during drinking bouts (*LeDue et al., 2015*; *Yapici et al., 2016*). Gustatory neurons located close to or inside the pharynx therefore seem to fulfill an important role in controlling the length of ingestion bouts by continuously surveying the quality of ingested food.

An intriguing finding from our study is that although sensillar and taste peg GRNs are crucial for specific aspects of the yeast feeding program, compromising either of these GRN subsets alone did not affect the total amount of yeast feeding. This highlights how the plasticity of feeding behavior renders these GRN subsets ultimately redundant in terms of total feeding. This finding is reminiscent of sweet taste in *Drosophila*, where although sensillar and pharyngeal GRNs play distinct roles in controlling the structure of sugar feeding behavior, these GRN subsets have been shown to be largely redundant in terms of total sugar intake (*LeDue et al., 2015*). Our study clearly shows that *Drosophila* is able to compensate for the perturbation of a specific set of sensory neurons, and the feeding subprogram they control, by altering other behavioral parameters underlying feeding such as the total number of feeding bursts. This parallel architecture, which employs distributed gustatory neurons in the proboscis of the fly, ensures sufficient uptake of critical nutrients through flexible behavioral programs.

Yeast is a complex, multimodal resource, and thus it is likely that multiple stimuli present in yeast activate *Drosophila* chemosensory neurons (*Fischler et al., 2007*; *Hussain et al., 2016b*; *Wisotsky et al., 2011*). At long ranges, flies are attracted by yeast volatiles detected through olfactory receptor neurons (*Becher et al., 2012*; *Christiaens et al., 2014*; *Dweck et al., 2015*). Olfaction is also important for efficient recognition of yeast as a food source (*Corrales-Carvajal et al., 2016*). However, we show that input from the gustatory system is ultimately critical to drive feeding on yeast. Multiple yeast fermentation products, including carbonation, activate the gustatory neurons identified in this study. However, these stimuli on their own do not induce a feeding rate that approximates the phagostimulatory power of yeast. Rather, our results suggest that multiple yeast stimuli must coincide to produce a yeast percept, and that yeast GRNs are likely to be specialized to respond to a variety of chemicals normally found in this microorganism. These chemicals are likely to include both those we showed to activate subsets of *Ir76b* GRNs, and other yeast metabolites currently not known to be detected by *Drosophila*. This would allow flies to ensure the reliable

detection of this essential food source, while permitting selective regulation of feeding on yeast depending on internal state. The emerging picture is that multiple yeast metabolites are detected by different gustatory neurons on the proboscis which act redundantly to mediate yeast intake, thus forming a proxy for the perception and ingestion of protein-, and therefore AA-, rich food.

Intriguingly, some *Ir76b* + GRNs in the legs have been shown to respond to AAs (*Ganguly et al., 2017*). However, we show here that tarsal GRNs are dispensable for yeast feeding, and do not contribute to the *Ir76b-GAL4* silencing phenotype. Furthermore, *Ir20a*, the putative receptor that conveys AA responsiveness, is not expressed in labellar GRNs (*Koh et al., 2014*), and flies lacking *Ir76b*, which is required for these AA responses, feed on yeast to the same extent as controls. Additionally, AAs hardly activate *Ir76b* GRNs on the labellum, suggesting that other components of yeast mediate this response. It is therefore possible that AA taste does not significantly contribute to the detection of yeast but that flies are also able to detect AAs independently to ensure their uptake from non-yeast sources. Furthermore, it is interesting to note that *Ir76b* GRNs have been shown to mediate the taste of substances which are important to support reproduction (*Ganguly et al., 2017*; *Hussain et al., 2016b*; *Walker et al., 2015*; *Zhang et al., 2013*). *Ir76b* GRNs could therefore be specialized in mediating the intake of foods that are relevant for reproduction.

Overall, our data demonstrate that deprivation from a particular nutrient – AAs – specifically increases the gain of gustatory neurons detecting the ethological food substrate – yeast – that provides the animal with this nutrient (*Figure 7*). In principle, such nutrient-specific sensory gain modulation could represent a general mechanism through which internal states could change the salience of a resource by directing behaviors towards stimuli that are most relevant in a specific state.

## Materials and methods

### Fly husbandry

All data are from yeast-deprived mated female flies unless otherwise stated. Flies were reared at 18 or 25°C, 70% relative humidity on a 12 hr light-dark cycle. Experimental and control flies were reared at standard density and were matched for age and husbandry conditions. The fly medium (yeast-based medium [YBM]) contained, per litre, 80 g cane molasses, 22 g sugar beet syrup, 8 g agar, 80 g corn flour, 10 g soya flour, 18 g yeast extract, 8 ml propionic acid, and 12 ml nipagin (15% in ethanol). For experiments that did not involve thermogenetic silencing (including calcium imaging and optogenetic inhibition), flies were kept at 25°C. Yeast deprivation was induced by feeding flies for 3 (or 10) days on a tissue soaked with 6.5 ml of 100 mM sucrose. For nutrient-specific deprivation, flies were reared on yeast-based food, transferred to fresh YBM for one day and then kept for 3 days on holidic medium with or without amino acids. This holidic medium was prepared as detailed in (*Corrales-Carvajal et al., 2016*; *Piper et al., 2014*), using the 50S200NYaa composition to approximate the amino acid ratio found in yeast. For experiments involving thermogenetic silencing, flies were reared and kept at 18°C. Yeast deprivation was induced by keeping flies for 7 days on a tissue soaked with 100 mM sucrose. This longer deprivation time was chosen to compensate for the lower metabolic rate at colder temperatures.

### Drosophila stocks and genetics

Neuronal silencing was achieved using the *20xUAS-shibire^ts* transgene inserted in the *VK00005* or *attP5* landing site (gift of Gerry Rubin, HHMI Janelia Research Campus). Genetic backgrounds of the control animals were matched as closely as possible to the experimental animals. For silencing experiments, the *VK00005* or *attP5* landing site background was crossed to the *GAL4* line as a control. The *Poxn* mutant fly experiments contained the following genetic manipulations (gift of Markus Noll and Werner Boll): homozygous control: *w^1118*; heterozygous control: *w^1118*; *Poxn^ΔM22-B5*/+; homozygous mutant: *w^1118*; *Poxn^ΔM22-B5* homozygote with *ΔSfoBs105/ΔSfoBs127* to rescue all defects other than taste sensilla. The *atonal* mutant fly experiments relied on the following genotypes (gift of Ilona Kadow): Control flies (*ato+/+*): *eyflp; FRT82B/FRT82B, CL*; Flies with *atonal* mutant antennae (*ato-/-*): *eyflp; FRT82B ato[1]/FRT82B, CL*. The full genotypes of the lines used in the manuscript are listed in *Supplementary file 1*.

The *Poxn-GAL80* vector was a gift of Duda Kvitsiani and Barry Dickson. Briefly, the 14 kb *Poxn* enhancer fragment from the Poxn-GAL4-14 vector (*Boll and Noll, 2002*) was cloned into a custom GAL80-containing vector, which was injected into $w^{1118}$ embryos.

To generate *Ir76b-GAL80*, we amplified the enhancer fragment of *Ir76b* using the following primers: 5'-CCCAGTCTAATGTATGTAATTGCC, 5'-CGATACGAGTGCCTACTGTAC, and cloned it into the pBPGAL80Uw-6 vector (AddGene). This vector was separately inserted into the attp40 and attP2 sites by PhiC31 integrase-mediated recombination. Injections were performed by BestGene.

## Thermogenetic neuronal silencing experiments

Flies carrying the temperature-sensitive allele of *shibire* under UAS control were crossed with a collection of different *GAL4* lines in order to acutely silence distinct neuronal subpopulations in experimental flies. Flies were reared at 18°C. 7–10 days after eclosion, female flies were sorted into fresh YBM and *Canton-S* males were added to ensure mating. Two days later, flies were transferred again to YBM; on the following day, they were transferred to 100 mM sucrose solution for 7 days to induce a yeast deprivation state. Silencing was induced by preincubating flies at 30°C for two hours and performing the two-color food choice or the flyPAD assays at this temperature. Control flies were always kept at 18°C, including during the assay.

## Two-color food choice assays

Two-color food choice assays were performed as previously described (*Ribeiro and Dickson, 2010*). Groups of 16 female and five male flies were briefly anesthetized using light $CO_2$ exposure and introduced into tight-fit lid petri dishes (Falcon, Corning, NY, USA #351006). For the yeast choice assays, the flies were given the choice between nine 10 µl sucrose spots mixed with red colorant (20 mM sucrose [Sigma-Aldrich, St Louis, MO, USA, #84097]; 7.5 mg/ml agarose [Invitrogen, Waltham, MA, USA, #16500]; 5 mg/ml Erythrosin B [Sigma-Aldrich, #198269]; 10% PBS) and nine 10 µl spots of yeast mixed with blue colorant (10% yeast [SAF-instant, Lesaffre, France]; 7.5 mg/ml agarose; 0.25 mg/ml Indigo carmine [Sigma-Aldrich, #131164]; 10% PBS) for 2 hr at 18, 25 or 30°C, 70% RH, depending on the experimental condition. Flies were then frozen and females scored by visual inspection as having eaten either sucrose (red abdomen), yeast (blue abdomen), or both (red and blue or purple abdomen) media. The yeast preference index (YPI) for the whole female population in the assay was calculated as follows: $(n_{blue\ yeast} - n_{red\ sucrose})/(n_{red\ sucrose} + n_{blue\ yeast} + n_{both})$. For amino acid preference experiments, yeast was replaced with a holidic diet (200NHunt) lacking sucrose, and sucrose was replaced with a holidic diet (50 mM sucrose) lacking amino acids, as described in (*Leitão-Gonçalves et al., 2017*). The holidic diet was prepared as detailed in (*Corrales-Carvajal et al., 2016*; *Piper et al., 2014*) according to the Hunt amino acid ratio. Both solutions were in 1% agarose. Values for *n* shown in the figures indicate the number of food choice assays performed.

## flyPAD assays

flyPAD assays were performed based on a protocol previously described (*Itskov et al., 2014*). One well of the flyPAD was filled with 20 mM sucrose, and the other with 10% yeast, each in 1% agarose.

In *Figure 1—figure supplements 1A* and *3B*, only one well of the flyPAD was filled with the stimulus solution in 1% agarose. Deactivated yeast was generated using an autoclave; carbonation was produced by mixing 0.2 ml of 1 M $NaHCO_3$ with 0.8 ml of 1 M $NaH_2PO_4$ immediately before use (*Fischler et al., 2007*); glycerol (Sigma-Aldrich, #G6279) was used at 10 mM (approximately the concentration in live yeast [*Scanes et al., 2017*]) and putrescine (Sigma-Aldrich, #51799) at 5 mM. AA solutions were prepared as detailed in (*Corrales-Carvajal et al., 2016*; *Piper et al., 2014*) according to the Hunt amino acid ratio, with and without the rest of the holidic diet components (excluding sucrose).

Flies were individually transferred to flyPAD arenas by mouth aspiration and allowed to feed for one hour at the indicated temperature, 70% RH; except for *Figure 1—figure supplement 1A* and *Figure 1—figure supplement 3B*, where flies fed for 30 min. flyPAD data were acquired using the Bonsai framework (*Lopes et al., 2015*), and analyzed in MATLAB using custom-written software, as described in (*Itskov et al., 2014*). Values for *n* shown in the figures indicate the number of flies tested.

## Optogenetic neuronal silencing experiments

For closed-loop optogenetic silencing experiments using the flyPAD setup, flies expressing *20xUAS-GtACR1* (*Mohammad et al., 2017*) were deprived of yeast for 3 days on a tissue soaked with 6.5 ml of 100 mM sucrose, 400 µM all-*trans*-retinal (Sigma-Aldrich, #R2500, using a stock solution of 100 mM all-*trans*-retinal in ethanol). Experiments were performed at 25°C, 70% RH. The flyPAD was modified by adding multi-color high-power (10W) RGBA LEDs (LED Engin, San Jose, CA, USA) above each arena. Both channels were filled with 10% yeast mixed with 1% agarose. Periods of active interaction of the fly with the food (activity bouts [*Itskov et al., 2014*]) were detected from each capacitance channel in real time using a custom Bonsai workflow (*Lopes et al., 2015*). For the stimulated 'experimental' channel, detection of an activity bout was programmed to send an activation pulse to the green (523 nm) LED if the bout was sustained for 250 ms. Measurements using a photodiode identified a latency of 86.5 ms (mean with a range of 50–120 ms) between sending the activation pulse and the LED being illuminated. Thus, the total latency from activity bout onset to LED activation was 300–370 ms. This latency was chosen such that the LED would be activated only after feeding burst initiation: in $w^{1118}$ females deprived of yeast for 3 days, >95% of activity bouts on a yeast patch led to feeding burst initiation within 300 ms. Since flies sip at a rate of around 5 Hz within feeding bursts (*Itskov et al., 2014*), LED activation would occur consistently within the first 1–2 sips in an activity bout. The LED activation was sustained for 1.5 s independently of the activity of the fly. After this, the closed-loop optogenetic workflow would begin again. For the unstimulated 'control' channel, interaction of the fly with the food did not trigger LED activation. We used high-speed video analysis to ensure that optogenetic inhibition of taste peg GRNs using this protocol did not lead to the unspecific termination of feeding by triggering startle responses or other obvious behavioral artifacts.

## Proboscis extension response assays

PER assays were performed as described in (*Walker et al., 2015*). Mated female flies of the indicated genotypes were deprived of yeast for 3 days as described above, and were then gently anesthetized with $CO_2$, and affixed by the dorsal thorax onto a glass slide using No More Nails (Unibond) in groups of 15–20 flies. Flies were allowed to recover for 2 hr at 25°C in a humidified box, and were then moved to room temperature (~22°C), where the behavioral experiments were performed under a dissection microscope. Flies were first allowed to drink water until they no longer responded to a 5 s stimulation, and then a droplet of 10% yeast was presented for 3 s on the labellum. Flies were scored as 1 (full extension), 0.5 (partial extension), or 0 (no extension), and each fly was treated as a single data point for each stimulus.

## Immunohistochemistry, image acquisition and 3D rendering

Males from each *GAL4* line were crossed to females homozygous for the *UAS-CD8::GFP* reporter line and 3–10 day-old adult females carrying both the *GAL4* driver and the *UAS* reporter were dissected. Samples were dissected in 4°C PBS and were then transferred to formaldehyde solution (4% paraformaldehyde in PBS + 10% Triton-X) and incubated for 20–30 min at RT. Samples were washed three times in PBST (0.3% TritonX in PBS) and then blocked in Normal Goat Serum 10% in PBST for 2–4 hr at RT. Samples were then incubated in primary antibody solutions (Rabbit anti-GFP [Torrey Pines Biolabs, Secaucus, NJ, USA] at 1:2000 and Mouse anti-NC82 [Developmental Studies Hybridoma Bank] at 1:10 in 5% Normal Goat Serum in PBST). Primary antibody incubations were performed overnight at 4°C with rocking. They were then washed in PBST 2–3 times for 10–15 min at RT and again washed overnight at 4°C. The secondary antibodies were applied (Anti-mouse A594 [Invitrogen] at 1:500 and Anti-rabbit A488 [Invitrogen] at 1:500 in 5% Normal Goat Serum in PBST) and brains were then incubated for 3 days at 4°C. They were again washed in PBST 2–3 times for 10–15 min at RT, and washed overnight at 4°C. Before mounting the samples were washed for 5–10 min in PBS. Samples were mounted in Vectashield (Vector Laboratories, Burlingame, CA, USA). Images were captured on a Zeiss LSM 710 using 10x or 20x objectives. Images of the periphery did not include immunostainings. Heads and legs of female flies were clipped off and placed in a drop of Oil 10S (VWR chemicals, Center Valley, PA, USA) between a slide and a cover slip. Images were captured on a Zeiss LSM 710 using a 10x objective. Note that during the mounting procedure of the fly head, the labellum opens, exposing the inner surface of the labellum and the taste pegs.

3D reconstructions of the nervous system and the periphery were generated using FluoRender (*Wan et al., 2009*; *Wan et al., 2012*).

## Calcium imaging

The preparation for calcium imaging was adapted from that described in (*Flood et al., 2013*). Each fly was lightly anesthetized using $CO_2$, and fixed into a custom-made chamber using UV-curing glue (Bondic, Aurora, ON, USA). The proboscis was fixed by the rostrum in an extended position, and UV-curing glue was used to seal around the head capsule within the imaging window, which was then immersed in AHL saline (103 mM NaCl, 3 mM KCl, 5 mM TES, 10 mM trehalose dihydrate, 10 mM glucose, 2 mM sucrose, 26 mM $NaHCO_3$, 2 mM $CaCl_2$ dihydrate, 4 mM $MgCl_2$ hexahydrate, 1 mM $NaH_2PO_4$, pH7.3) bubbled with 95%$O_2$/5%$CO_2$. A 30G needle (BD Microlance 3, Becton Dickinson, Franklin Lakes, NJ, USA) was used to cut the cuticle along the edges of the eyes, just above the rostrum and just below the ocelli, and this piece of cuticle along with the antennae was removed using forceps. Air sacs and fatty tissue surrounding the SEZ were removed, but the esophagus was left intact. The stage was then transferred to a mount under a two-photon resonant-scanning microscope (Scientifica, UK) equipped with a 20x NA 1.0 water immersion objective (Olympus, Japan). During imaging, the brain was constantly perfused with AHL saline bubbled with 95%$O_2$/5%$CO_2$.

A 920 nm laser (Coherent, Santa Clara, CA, USA) was used to excite GCaMP6s through a resonant scanner, and emitted fluorescence was recorded using a photomultiplier tube. The objective was controlled by a piezo-electric z-focus, allowing serial volumetric scans of the SEZ. The SEZ was imaged at 31 z-positions, with the upper and lower limits defined in order to encompass the entire SEZ (~60–80 µm), at a frame rate of 61.88 Hz, with a 235 × 117 µm (512 × 256 pixel) frame. Imaging data were acquired using SciScan (Scientifica). A frontal view of the fly, illuminated by scattered light from the laser, was simultaneously acquired through a PointGrey Flea3 camera using Bonsai. Each trial consisted of 4650 frames (150 volumes); the gustatory stimulus was applied to the labellum at approximately volume #51 and removed at volume #100 using a pulled glass micropipette filled with the stimulus solution mounted on a micromanipulator (Sensapex, Finland). Each stimulus was applied 2–3 times per fly, and the mean response per fly used for further analysis. For comparing responses across dietary conditions, imaging sessions using flies of each condition were interleaved. Stimuli were dissolved or suspended in water as follows: 10% w/v SAF-INSTANT yeast; 10% yeast deactivated using an autoclave; 1% dry ice; 200 mM $NaHCO_3$ in 500 mM $NaH_2PO_4$ (mixed immediately before use); 10 and 100 mM glycerol; 10 and 100 mM putrescine; FLYAA amino acid ratio (without holidic base) at 200 and 300 mM total nitrogen (*Piper et al., 2017*); 20 and 500 mM sucrose.

## Calcium imaging analysis

All analysis of imaging data was performed using custom scripts in MATLAB (Mathworks, Natick, MA, USA). To facilitate analysis of imaging data, we took the average intensity projection of each serial volume stack (31 frames). A 5 × 5 pixel median filter was applied to each such projection, and each projection was registered to the first projection collected from that fly to correct for movement, using a discrete Fourier transform-based subpixel rigid registration algorithm (*Guizar-Sicairos et al., 2008*). For each fly, regions of interest (ROIs) were manually defined based on the maximum intensity projection of the first trial for this fly. Two-photon and camera images were aligned using the onset of scattered laser illumination in the camera image. Each trial was then aligned to stimulus onset based on manual annotation of stimulus application time. For each trial and ROI, the mean fluorescence intensity within the ROI at each time point was used as $F$, and $F_0$ calculated as the median of these values from 22 to 2 frames before the annotated stimulus onset. $\Delta F/F_0$ was calculated as $(F-F_0)/F_0$ for each time point. To calculate area under the curve, we took the sum of the $\Delta F/F_0$ values from 0 to 44 frames after stimulus onset for each fly's average trace, except for *Figure 4—figure supplement 1C and D*, for which frames 0 to 19 were used due to the shorter stimulation time. For each fly, we treated each brain hemisphere as a separate data point; values for $n$ shown in the figures indicate the number of flies imaged per condition.

## Statistics

Results of two-color food choice assays, flyPAD and calcium imaging experiments were compared using the Kruskal-Wallis test, followed by Dunn's multiple comparison test. For tests comparing only

two groups, the Wilcoxon rank-sum test was used. PER data were compared using $2 \times 2$ Fisher's exact tests, appropriate for categorical data; and 95% confidence intervals were calculated using the modified Wald method (*Agresti and Coull, 1998*). To analyze the effects of mating and protein deprivation, groups were compared by two-way ANOVA. All tests were two-tailed.

## Acknowledgements

We thank Richard Benton, Gerry Rubin, Kristin Scott, Leslie Vosshall, Ilona Grunwald-Kadow, Craig Montell, Scott Waddell, Ann-Shyn Chiang, Amita Sehgal, Aaron DiAntonio, Elizabeth Gavis, Michael Pankratz, Irene Miguel-Aliaga, Bader Al-Anzi, Ping Shen, Werner Boll, Barry Dickson, Sofia Lavista-Llanos, Adam Claridge-Chang, and Gero Miesenböck for providing fly strains. We also thank Ulrike Heberlein, Wes Grueber, the NP consortium, Douglas Armstrong, and numerous other researchers that have contributed to our collection of *GAL4* stocks. Further lines obtained from the Bloomington Drosophila Stock Center (NIH P40OD018537), and the VDRC (Vienna, Austria) were used in this study. The enhancer trap *GAL4* silencing screen was performed at the IMP in the laboratory of Barry J. Dickson. We thank Barry J. Dickson, Martin Häsemeyer, Nilay Yapici, Christoph Treiber, Lorenz Pammer, Carolina Doran, and Ana Paula Elias for assistance during the screen and post-hoc analysis. We thank Richard Benton, Juan Antonio Sánchez-Alcañiz and Kristin Scott for training and assistance in establishing our two-photon calcium imaging prep. We thank Eugenia Chiappe, Michael Orger, Dennis Goldschmidt, Daniel Münch, and members of the Behavior and Metabolism laboratory for helpful discussions and comments on the manuscript. We especially thank Yu-Chieh David Chen for showing us that the scientific system is changing for the better by providing excellent feedback on our preprint deposited on bioRxiv. Finally, we thank Gil Costa for assistance with illustrations. This project was supported by the Portuguese Foundation for Science and Technology (FCT) grant PTDC/BIA-BCM/118684/2010, and postdoctoral fellowship SFRH/BPD/79325/2011 to PMI; the Human Frontier Science Program Project Grant RGP0022/2012; the BIAL Foundation Grants (283/14 and 279/16); and the Marie Curie FP7 Programme FLiACT (ITN) grant. Research at the Centre for the Unknown is supported by the Champalimaud Foundation.

## Additional information

### Competing interests

Pavel M Itskov: PMI has a commercial interest in the flyPAD open-source technology. The other authors declare that no competing interests exist.

### Funding

| Funder | Grant reference number | Author |
|---|---|---|
| Fundação para a Ciência e a Tecnologia | PTDC/BIA-BCM/118684/2010 | Carlos Ribeiro |
| Fundação para a Ciência e a Tecnologia | SFRH/BPD/79325/2011 | Pavel M Itskov |
| Human Frontier Science Program | RGP0022/2012 | Carlos Ribeiro |
| Fundação Bial | 283/14 | Carlos Ribeiro |
| Fundação Bial | 279/16 | Carlos Ribeiro |
| European Commission | FLiACT ITN | Carlos Ribeiro |
| Champalimaud Foundation | | Carlos Ribeiro |

The funders had no role in study design, data collection and interpretation, or the decision to submit the work for publication.

## Author contributions
Kathrin Steck, Conceptualization, Data curation, Formal analysis, Investigation, Visualization, Methodology, Writing—original draft, Writing—review and editing; Samuel J Walker, Conceptualization, Data curation, Software, Formal analysis, Investigation, Visualization, Methodology, Writing—original draft, Writing—review and editing; Pavel M Itskov, Software, Formal analysis; Célia Baltazar, Formal analysis, Investigation; José-Maria Moreira, Software; Carlos Ribeiro, Conceptualization, Data curation, Formal analysis, Supervision, Funding acquisition, Investigation, Methodology, Writing—original draft, Project administration, Writing—review and editing

## Author ORCIDs
Kathrin Steck (ID) http://orcid.org/0000-0003-2711-2873
Samuel J Walker (ID) http://orcid.org/0000-0003-3118-8467
Carlos Ribeiro (ID) http://orcid.org/0000-0002-9542-7335

## Decision letter and Author response
Decision letter https://doi.org/10.7554/eLife.31625.024
Author response https://doi.org/10.7554/eLife.31625.025

## Additional files
### Supplementary files
• Supplementary file 1. Supplementary Tables 1-3.
DOI: https://doi.org/10.7554/eLife.31625.021

• Transparent reporting form
DOI: https://doi.org/10.7554/eLife.31625.022

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
