## [Decision Letter]

Thank you for submitting your work entitled "Internal amino acid state modulates yeast taste neurons to support protein homeostasis in *Drosophila*" for consideration by *eLife*. Your article has been reviewed by three peer reviewers, and the evaluation has been overseen by Mani Ramaswami as the Reviewing Editor and K VijayRaghavan as the Senior Editor. The reviewers have opted to remain anonymous.

Our decision has been reached after consultation between the reviewers. Based on these discussions and the individual reviews below, we regret to inform you that your current manuscript cannot be considered further for publication in *eLife*.

The main reason for this decision is the consensus view that the GRNs involved as well as their contribution to the behavioral response to yeast have not yet been established with sufficient precision. The reviewers agreed that the use of intersectional strategies suggested by your work (e.g. between *Ir76b* and *Ir25* or *Ir76b* and *1261-Gal4*) or perhaps even targeted laser lesioning of suspected neurons, when combined with your existing and potentially modified behaviour tests, had the potential to greatly add to the strength of your interesting and significant conclusions. However, they also agreed that performing these essential experimental additions would take well beyond the 2-months that *eLife* allows for resubmission.

Given the overall general enthusiasm and recommendation of the reviewers, the journal is, however, willing to consider a new submission from you that addresses the key reviewer concerns, if you choose to follow this route. If you are confident that you can address the principal concerns within two months, we will then be able to consider a revised submission speedily.

*Reviewer #1:*

This manuscript by Steck et al. describes functionally redundant subsets of GRNs on the proboscis essential for normal feeding response to yeast. Imaging analyses have revealed these distributed GRNs displayed excitatory responses to the taste of yeast. They also provide convincing evidence that the internal AA state of the fly selectively modulates yeast GRNs. Another interesting finding is that reproductive state, which modulates yeast feeding behavior, dose not modulate yeast GRNs. Their findings have provided fresh insights into a circuit mechanism for homeostatic regulation of protein intake. Overall, the data appear to support their main conclusions. Given that circuit mechanisms underlying the homeostatic control of protein intake remain poorly understood, this work is interesting and significant. I believe it will be appropriate for publication in *eLife* after some modifications and editing.

1) Figure 1—figure supplement 3: In panel A *Ir25a*>*shi^ts1^* and *Ir76d*>*shi^ts1^* flies seem to prefer HB+suc food at 18 °C, while *1261*> *shi^ts1^* flies showed no such bias. Could that be due to the overexpression of functional *shi^ts1^* in *Ir25a* and *Ir76d* neurons? Any explanations?

Figure 3, were flies food or yeast-deprived? If so, for how long? This is especially important given that these neurons are sensitively modulated by food deprivation.

Figure 5—figure supplement 1: the color scheme is confusing.

*Reviewer #2:*

The mechanisms underlying the homeostatic regulation of protein intake are poorly understood. Both peripheral and central mechanisms are important in this process, and here, Steck et al. seek to characterize the former in *Drosophila*. From a screen of enhancer trap lines, they identify Gal4 drivers that are required for yeast intake and show that GRNs labeled by these drivers respond to specific protein hunger-inducing inputs (amino acid deprivation vs. mating status). Similar phenomenological data have previously been shown in locusts (protein deprivation increases activity of protein-related GRNs, but not sugar-related GRNs). However, these observations by Steck et al. are important because elucidating circuit mechanisms in *Drosophila* lays the groundwork for an in-depth dissection of the mechanisms underlying protein hunger, including how peripheral and central mechanisms are integrated to mediate this process. In general, the work is interesting, carefully performed, and clearly explained. However, there are a number of issues that should be addressed.

1) One of the main findings of the paper relates to the identification of the GRNs required for protein feeding. However, the identity of these GRNs is only indirectly assessed using *Ir76b-Gal80*. It would be much better to use positive intersectional approaches (e.g., split-*Gal4*) to specifically reveal and manipulate these GRNs.

2) The authors show that individually silencing taste sensilla, taste pegs, and pharyngeal taste organs does not significantly affect yeast feeding. They conclude that the proboscis GRNs in these different taste structures are functionally redundant. However, this is implied, not directly shown. Moreover, because the authors are assaying feeding, it may instead be that the "redundancy" may instead exist with central mechanisms. To address this issue, the authors should elicit the proboscis extension reflex by applying the food to the proboscis (rather than the legs). Related to this point, Masek et al. (PLOS Genetics 2013) showed that a fly's response to yeast (as measured by PER) was eliminated in Poxn mutants. Because the authors do not clearly isolate the protein-related GRNs as mentioned above, it remains possible that central brain neurons are being silenced when using the different Gal4 drivers.

3) Another finding that differs from prior published work: Ganguly et al., 2017) found that *Ir76b* mutants exhibit substantially reduced yeast preference, whereas the authors find that *Ir76b* mutants have intact yeast preference in protein-deprived animals. How do the authors reconcile these findings?

*Reviewer #3:*

In the manuscript by Steck et al., the authors identify gustatory receptor neurons (GRNs) of the vinegar fly, *Drosophila melanogaster*, that are critical for yeast preference, and show that these neurons transduce a signal that is modulated by the amino acid deprivation. The authors first show that when flies are deprived of amino acids for several days, they increase specific desire for yeast. The authors use a series of genetic expression experiments to show that three gal4 lines (IR76B, IR25A, and *1261-Gal4*) all label a subset of neurons in the primary taste neurons critical for the increased desire. They went on to show that each driver labels taste pegs, sensilla, and pharyngeal GRNs, but none of these three components on their own were sufficient to eliminate yeast preference, suggesting redundancy. They go on to show that the AMS1 and PMS4 regions, which receive input from the taste neurons and taste sensilla, respectively, respond to yeast and associated components (glycerol, carbonation and putrescine) with calcium elevation. Additionally, they showed that IR76B GRN-derived calcium response to yeast was strongly elevated after deprivation, whereas sugar GRN response was attenuated. They also showed that this deprivation was linked specifically to amino acids in diet, whose absence modulates gain. Finally, they show that mating increases yeast appetite, but not strength of GRN calcium response, suggesting that mating does not affect this circuit, and instead drives increased yeast appetite through other means.

Overall, I enjoyed reading the manuscript and I find the results very interesting. The paper is a comprehensive study of a circuit that has received a good deal of attention but in its current form the manuscript has many gaps and fail to pin point the GRNs that are responsible for yeast taste. Also, I think it is especially important to clarify the results of the pharyngeal GRN silencing, especially *Ir94f-Gal4*. There is also no gain of function experiment suggesting taste is the only driving factor that drives yeast preference as strongly suggested in the manuscript. Optogenetic activation of GRNs that show to regulate yeast preference will complement the silencing experiments. After reading the manuscript I still don't know exactly which GRNs are responsible for yeast taste and where they project in SEZ; there is a disconnection between the silencing and imaging experiments.

1) In Figure 1 and Figure 1—figure supplement 3, authors argue that amino acids are not the sole stimuli mediating yeast appetite. I would be curious to see how the flies will respond to yeast without amino acids. These yeast sources are commercially available. In the manuscript, major components of yeast taste (glycerol, carbonation, amino acids, etc.) are shown to be not required for yeast appetite, however authors do not suggest or provide a clue about what yeast component is important for flies to taste yeast. I understand this is hard but at least some suggestions would be good to mention in the Discussion.

2) In Figure 2, authors use the *poxn-GAL80* to suppress the expression in GRNs. Some of the *Poxn* enhancers have been shown to label central neurons in the fly brain (see The *Drosophila* Pox neuro gene: control of male courtship behavior and fertility as revealed by a complete dissection of all enhancers, Werner Boll, Markus Noll, Development). It would be good to include the expression pattern of the *Poxn-4-14-Gal4* or provide evidence that *Poxn-Gal80* only suppresses expression of GRNs. In Figure 2 see GFP expression in *IR76b-Gal4* outside of chemosensory regions in the fly brain. This expression is suppressed with *Poxn-Gal80*. There is a possibility that internal yeast sensing might be involved in yeast feeding, thus it would be good to clarify the expression of the enhancer that is used to generate the Gal80. Also in Figure 2, there are still GFP positive neurons in VNC. What is the evidence that these neurons are not involved in yeast preference? The authors claim that yeast preference is due to only yeast taste and this sentence is not completely accurate.

3) In Figure 2—figure supplement 3 see a lot of variation in different Gr and IR silencing experiments, some of which are significant. Could authors comment on this? Especially *Ir60b-Gal4* and *Ir94f-Gal4* seem to have an impact on yeast preference. There is a clear difference with *Ir94f-Gal4*>*shi^ts^* at 30 °C vs. 18 °C. However, because authors do not show how controls behave in 18 °C, it is inconclusive. Why are authors not mentioning these results in the text? These two Gal4s have been shown to regulate sugar ingestion and might be involved in yeast appetite too (see A receptor and neuron that activate a circuit limiting sucrose consumption, Joseph R et al. *ELife*, 2017). This paper also has not been cited in the references.

4) The authors test AMS1 and PMS4 calcium response to carbonation, glycerol, and putrescine, but did not test mixed amino acids, a critical component of the yeast diet (Figure 3—figure supplement 1).

---

## [Author Response]

[Editors’ note: The authors returned a revised manuscript within the two-month timeframe. The authors’ response to the reviewers’ concerns follows.]

[…] The main reason for this decision is the consensus view that the GRNs involved as well as their contribution to the behavioral response to yeast have not yet been established with sufficient precision. The reviewers agreed that the use of intersectional strategies suggested by your work (e.g. between Ir76b and Ir25 or Ir76b and 1261-Gal4) or perhaps even targeted laser lesioning of suspected neurons, when combined with your existing and potentially modified behaviour tests, had the potential to greatly add to the strength of your interesting and significant conclusions. However, they also agreed that performing these essential experimental additions would take well beyond the 2-months that eLife allows for resubmission.Given the overall general enthusiasm and recommendation of the reviewers, the journal is, however, willing to consider a new submission from you that addresses the key reviewer concerns, if you choose to follow this route

We thank the editors and the reviewers for their general enthusiasm. We focused on addressing all the points raised by the reviewers and we are confident that we managed to satisfy all requests.

We agree that identifying specific subsets of neurons would be an exciting addition to the manuscript. We therefore focused on this key point in our efforts to revise the manuscript and we are happy to report that we can now show that a specific subset of *1261-GAL4/Ir76b/Ir25a* gustatory neurons, namely the taste peg neurons, fulfill an important function in sustaining yeast feeding. These new data therefore assign a specific functional role in yeast feeding to a subset of the neurons previously identified in our study. To our knowledge, this is the first time a specific function in mediating feeding has been assigned to this group of taste neurons. We would also like to highlight that we use a state-of-the-art approach to prove the involvement of taste peg gustatory neurons in yeast feeding. This approach relies on closed-loop optogenetic inactivation of neurons specifically after the initiation of feeding bursts. Furthermore, in the revised manuscript, we also use the proboscis extension response assay, a classic gustatory assay, to show that sensillar gustatory neurons are likely to be involved in initiating yeast feeding, as flies without functional gustatory sensilla do not show proboscis extension towards yeast.

Importantly, our original conclusion still remains valid. Despite the specific function of taste pegs and taste sensillar gustatory neurons in mediating specific aspects of yeast feeding, the total level of yeast feeding remains unchanged when either of these two subsets of neurons is inactivated. Therefore, our original conclusion that the different sets of gustatory neurons are likely to act redundantly to ensure robust feeding is supported by these new data. We now also provide behavioral evidence for our hypothesis that upon silencing of specific gustatory subsets, flies modify their feeding motor program to compensate for a potential reduction in total yeast feeding.

Addressing more specifically the points made above.

We still think that our original conclusion regarding yeast GRN organization is correct – namely that multiple subsets of gustatory neurons, all of which are common to the three lines we identified, coordinately mediate total yeast feeding. This conclusion is supported by three main arguments: 1) All the lines which we identified in the screen as abolishing yeast feeding show a broad and common overlap in specific neurons in the gustatory system, but not in other neurons (for example, these lines label different subsets of olfactory receptor neurons). 2) All neurons on the external surface of the labellum which we surveyed with our imaging setup show robust and strong responses to yeast taste, as well as to some yeast metabolic byproducts such as carbonation. Furthermore, the calcium response to yeast stimulation in the sensillar as well as peg GRN projections is regulated by the internal amino acid state of the animal, reflecting the regulation of yeast feeding behavior. 3) As demonstrated by our new data, silencing subsets of these neurons does not reduce the total intake of yeast, but does compromise specific aspects of the yeast feeding program. These new data show that taste peg neurons and very likely also sensillar GRNs participate in controlling distinct aspects of yeast feeding. However, silencing a specific subset leads to a compensatory change in the feeding program, which ultimately allows the fly to achieve the same level of total food intake as controls. This compensatory mechanism is very likely to rely on the function of the other gustatory neurons labeled in the lines we identified, as silencing all of these neurons prevents compensation and thereby almost completely abolishes yeast feeding.

We do however agree with the reviewers that identifying subsets of gustatory neurons controlling yeast feeding would significantly add to the current manuscript. As also suggested by the reviewers we have spent multiple years trying different intersectional strategies to reduce the number of neurons which are required to maintain yeast feeding. During these last two months, we redoubled our effort and performed additional intersections using further strategies but likewise, none of these have been successful. The strategies used over the last years relied on using GAL80 lines, FLP>FRT positive and negative intersections, and/or LexA-based positive and negative intersections. The results fell into two categories: either no experiment led to a reduction in total yeast feeding, or when a significant reduction in yeast feeding was observed, subsequent neuroanatomical analyses showed that the intersection was only partially or not at all successful in restricting the expression pattern. In the latter cases, the intersection either led to clonal effects (some chemosensory neurons were still labeled) or only led to a partial suppression of the expression level. Importantly, none of the experimental results contradicted our previous or current results and interpretations.

Inspired by comments from the reviewers we now chose a different strategy to identify restricted subsets of neurons with specific roles in yeast feeding. We used GAL4 lines that label specific subsets of GRNs within our identified GAL4 lines, and analyzed the effect of silencing these lines on specific aspects of yeast feeding. The logic of this approach was that different GRNs might support different aspects of yeast feeding, but compensatory mechanisms might mask the effect of manipulating these neurons on total yeast intake. Indeed, we now find that silencing the taste peg gustatory neurons alone changes the length of feeding bursts. Using this strategy, we therefore identify a specific function for taste peg GRNs in yeast feeding. We use a novel closed-loop optogenetic silencing approach to further confirm this finding. This function is specific to taste pegs, as silencing using multiple specific GAL4 lines labeling taste peg GRNs, and no other taste receptor neurons, led to the same effect on yeast feeding bursts; whereas silencing other neurons involved in feeding and taste, such as Gr5a GRNs or IN1 neurons, did not affect the length of yeast feeding bursts. Finally, consistent with previous findings, the changes in feeding burst length did not affect the total number of sips, since flies compensated by increasing the total number of yeast feeding bursts.

We also use different manipulations of sensillar gustatory neurons to show that one of the functions of the sensillar GRNs on the labellum is to promote the extension of the proboscis upon stimulation with yeast. We therefore also show a new function for the sensillar GRNs on the labellum in yeast feeding. Despite the drastic effects on proboscis extension, abolishing sensillar GRN functions has also no discernable effect on total yeast intake, reinforcing once more our concept of functional redundancy.

Regarding the proposal to use laser ablation to reveal specific functions of gustatory neuron subsets, we are not aware that this has ever been done for studying the behavioral function of chemosensory neurons in *Drosophila*. We are aware that laser ablation has been used to dissect the function of the *fdg* neurons, but these are very few central neurons which are readily accessible in an imaging context like the one we employ. We also do not believe that laser lesioning of sensory neurons is feasible in a feeding paradigm. Lesioning a large number of neurons within the proboscis would damage the cuticle and therefore impair proboscis function and feeding behavior in general. Furthermore, to lesion the axons of these neurons within the head cavity would require opening the head cavity in a head-fixed preparation. In our hands, the behavior of flies in such a head-fixed preparation is not comparable to that of freely-moving flies. Finally, this approach would not allow us to selectively ablate peg or sensillar GRNs, since their axons are intermingled in the labial nerve. Given these technical constraints, together with the fact that we now dissect the function of specific gustatory neuron subsets using genetic approaches, we decided not to pursue this proposed approach.

Reviewer #1:[…] 1) Figure 1—figure supplement 3: In panel A Ir25a>shi^ts1^ and Ir76d>shi^ts1^ flies seem to prefer HB+suc food at 18 °C, while 1261> shi^ts1^ flies showed no such bias. Could that be due to the overexpression of functional shi^ts1^ in Ir25a and Ir76d neurons? Any explanations?

This is an interesting point, and we agree that the partial phenotype at 18 °C in the experimental genotype is probably due to the strength of the drivers. We suppose that this is due to *Ir76b-GAL4* and *Ir25a-GAL4* lines being very strong drivers. Although the change from the permissive temperature to the restrictive temperature increases the probability of *Shibire* proteins being in the dominant-negative form, even at the permissive temperature there are always a few molecules in the dominant-negative form. Therefore, stronger drivers will increase the pool of dominant-negative forms in the targeted neurons, leading to a decrease in available synaptic vesicles. Importantly for our interpretation, the shift in temperature leads to a statistically significant decrease in amino acid preference with both drivers, and furthermore we do not see any effect at 18 °C with the somewhat weaker driver *1261-GAL4*.

Figure 3, were flies food or yeast-deprived? If so, for how long? This is especially important given that these neurons are sensitively modulated by food deprivation.

Yes, flies were yeast-deprived for 10 days. The pre-treatment of the flies used in these experiments, as well as all others, are described in the figure legends, as well as in the Materials and methods section.

Figure 5—figure supplement 1: the color scheme is confusing.

We apologize if the color scheme was confusing. We have amended the color scheme to match that of Figure 6 (previously Figure 5) and hope that it is clearer now.

Reviewer #2:[…] 1) One of the main findings of the paper relates to the identification of the GRNs required for protein feeding. However, the identity of these GRNs is only indirectly assessed using Ir76b-Gal80. It would be much better to use positive intersectional approaches (e.g., split-Gal4) to specifically reveal and manipulate these GRNs.

We agree that positive intersectional approaches are generally more informative than alternative approaches. We have indeed tried such approaches but none of these has been successful/informative for the reasons explained above. However, it is important to note that in this case the following variables make this approach less appropriate for our study: Given the available literature and our own neuroanatomical analyses, the *Ir76b-GAL4* and *Ir25a-GAL4* gustatory neuron populations are very likely to largely overlap (see Figure 1—figure supplement 2). Intersecting these two lines is therefore very unlikely to substantially reduce the number of labeled gustatory neurons in the *Ir76b-GAL4* line. Furthermore, the *1261-GAL4* line is an enhancer-trap line, and although we have mapped the genomic insertion site of the line and tried different approaches to identify an enhancer element reproducing the same expression pattern, we have not been successful in doing so. We therefore currently lack a large repertoire of genetic tools to do positive intersectional genetics using the *1261-GAL4* line.

Our plan was therefore to use lines which label subsets of gustatory neurons labeled by the identified lines to perform positive intersections. But conceptually this would be the same as directly using these lines to manipulate the specific neuronal subsets. Importantly, in the current manuscript, using careful behavioral analysis of yeast feeding behavior, we are able to show a function of specific subpopulations of GRNs in specific aspects of yeast feeding. We describe the details of this finding in the answer to the next point, but it is relevant for this point of the discussion that with this finding we can now pinpoint the importance of a specific subset of yeast taste neurons in mediating yeast feeding.

2) The authors show that individually silencing taste sensilla, taste pegs, and pharyngeal taste organs does not significantly affect yeast feeding. They conclude that the proboscis GRNs in these different taste structures are functionally redundant. However, this is implied, not directly shown. Moreover, because the authors are assaying feeding, it may instead be that the "redundancy" may instead exist with central mechanisms. To address this issue, the authors should elicit the proboscis extension reflex by applying the food to the proboscis (rather than the legs). Related to this point, Masek et al. (PLOS Genetics 2013) showed that a fly's response to yeast (as measured by PER) was eliminated in Poxn mutants. Because the authors do not clearly isolate the protein-related GRNs as mentioned above, it remains possible that central brain neurons are being silenced when using the different Gal4 drivers.

We thank the reviewer for this excellent suggestion. Following this idea, we analyzed the effect of manipulating specific subsets of gustatory neurons on different aspects of the yeast feeding microstructure (e.g. initiation of feeding, sustaining feeding, reinitiating feeding) using the flyPAD system. We observed that silencing taste peg neurons using specific lines led to a reproducible, significant shortening of the length of yeast feeding bursts.

We followed up this observation using a new closed-loop optogenetic silencing approach. We used recently published inhibitory opsins to specifically inhibit taste peg neurons once yeast feeding has been initiated. This closed-loop silencing approach leads to a dramatic reduction in yeast feeding burst length, confirming the importance of taste peg neurons in maintaining yeast feeding once initiated. Importantly, silencing other neuronal populations using this approach does not affect yeast feeding, confirming the specificity of this phenotype.

As suggested by the reviewer, we have also assigned a function to the sensilla population on the labellum. Using the proboscis extension response assay (PER) we are able to show that, as proposed by Masek and colleagues, taste sensilla are crucial for proboscis extension, and therefore feeding initiation, in response to yeast, in this case presented to the labellum. This is demonstrated by our findings that PER to yeast is strongly reduced both in *Poxn* mutants, and when a large subset of sensillar neurons is silenced using *Poxn-GAL4*.

Overall, the observation that interfering with specific gustatory subsets affects specific aspects of the feeding program but not overall feeding levels clearly indicates that the nervous system is able to robustly compensate for the loss of a specific subset of gustatory inputs, which ordinarily mediates a specific subprogram of feeding, by “homeostatically” modulating other aspects of food intake. This is exemplified by our finding that upon silencing of taste peg GRNs, flies compensate the reduction in the length of feeding bursts by increasing the number of feeding bursts, which ultimately allows the fly to maintain wild-type levels of yeast feeding.

Regarding the possibility that central neurons labeled by the identified *GAL4* lines contribute to the silencing phenotype, we can exclude this possibility on the basis of the neuroanatomy of *Ir25a-GAL4* neurons. We apologize for not having made this point clearer, but we mention in the manuscript that this line only labels sensory neurons, and no cell bodies in the central nervous system. This is also visible in Figure 1 and Figure 1—figure supplement 2. We can therefore exclude the hypothesis that the silencing phenotype is due to silencing non-sensory (central) neurons. We have now highlighted this point better in the manuscript by amending the passage to read:

“Importantly, this phenotype was due to a sensory deficit, and not to an effect of silencing neurons in the central nervous system, since *Ir25a-GAL4* exclusively labels peripheral neurons (note absence of cell bodies in brains and VNC of *Ir25a-GAL4* animals in Figure 1 and Figure 1—figure supplement 2).”

3) Another finding that differs from prior published work: Ganguly et al., 2017) found that Ir76b mutants exhibit substantially reduced yeast preference, whereas the authors find that Ir76b mutants have intact yeast preference in protein-deprived animals. How do the authors reconcile these findings?

It is important to note here an important difference between our study and that of Ganguly and colleagues: while we monitor yeast feeding, Ganguly et al. use yeast extract in their experiments. Yeast extract has a much lower phagostimulatory power than yeast (see for example Leitão-Gonçalves et al., 2017 Figure 2, where protein-deprived flies eat less yeast extract than sucrose, which is something we never see in this or other manuscripts when using whole yeast). This is likely due to the fact that yeast extract contains only the water-soluble fraction of yeast, and therefore lacks many other important stimuli driving feeding on yeast. The discrepancy between our data and those of Ganguly and colleagues can be easily explained by supposing that the stimuli mediating feeding on yeast extract depend on *Ir76b*, while complete yeast also contains stimuli which are *Ir76b*-independent.

Reviewer #3:[…] Overall, I enjoyed reading the manuscript and I find the results very interesting. The paper is a comprehensive study of a circuit that has received a good deal of attention but in its current form the manuscript has many gaps and fail to pin point the GRNs that are responsible for yeast taste.

As discussed above in our comments to the editor and reviewer 2, we now provide evidence that taste peg neurons are specifically responsible for maintaining feeding on yeast. We thereby provide the identity of a specific type of neurons involved in yeast feeding. Importantly, the function of taste peg neurons in controlling the length of feeding bursts is novel. Furthermore, the alteration in feeding bursts upon silencing of taste peg GRNs does not translate into a change in overall yeast intake, as the fly compensates by altering other aspects of their feeding behavior, including the total number of feeding bursts. This explains the lack of overall yeast feeding phenotype when manipulating only this subset of taste neurons, and supports our original interpretation that very likely different yeast gustatory subsets can compensate for the loss of other subsets.

Also, I think it is especially important to clarify the results of the pharyngeal GRN silencing, especially Ir94f-Gal4.

We apologize for not having made our point clear in the previous version of Figure 2—figure supplement 3. In this version of the figure, we originally omitted all the additional control experiments we had done in order not to overcrowd the figure. We only showed all of the control genotypes for lines that showed a statistically significant difference between the 18 and 30 °C conditions displayed at the top of the figure. Indeed, in this figure *Ir94f-GAL4* and *Gr28a-GAL4* do show a phenotype when compared to the 18 °C control. However, what is not clear in the previous version of the figure is that when compared to the genetic background control experiments these two lines do not show a significantly different level of yeast feeding. The reason for the ambiguity is that the relevant controls were only displayed in the bottom panels of the figure. As was visible in this lower panel, the apparent silencing phenotype disappears when considering the genetic background controls. We have now modified the figure to include all control experiments in all panels, and hope that the argument for a lack of phenotype is now made more clearly. We have also edited the corresponding text to make clear that although individual comparisons may be significant, none of these pharyngeal lines shows a significant reduction in yeast feeding compared to all of the relevant controls:

“None of these lines showed a reduction in yeast feeding compared to all controls, suggesting that the tested pharyngeal GRNs are not essential to support total yeast feeding.”

There is also no gain of function experiment suggesting taste is the only driving factor that drives yeast preference as strongly suggested in the manuscript. Optogenetic activation of GRNs that show to regulate yeast preference will complement the silencing experiments.

We have made multiple attempts to induce an increase in feeding on a non-yeast substrate by activating yeast GRNs using different *GAL4* lines labeling all or subsets of yeast gustatory neurons. Our attempts to perform “gain-of-function” experiments were based on the use of optogenetic approaches (CsChrimson), thermogenetic approaches (TrpA1), a constitutively active sodium channel (NaChBac), and lacing food with capsaicin and expressing the capsaicin receptor (VR1) in the gustatory neurons of interest. Some of our manipulations of specific GRN subsets did lead to slight but statistically significant increases in total feeding or feeding burst length. However, these effects were not solidly reproducible and differed among lines labeling the same sensory neuron subsets. We therefore decided not to include these data in this manuscript, and assume that while these neurons are required for yeast feeding, their activation alone is not sufficient to induce robust feeding.

After reading the manuscript I still don't know exactly which GRNs are responsible for yeast taste and where they project in SEZ; there is a disconnection between the silencing and imaging experiments.

We hope that the new data in which we show the involvement of the peg neurons in controlling yeast feeding burst length satisfies the expectations of the reviewer about pinpointing a specific set of neurons mediating yeast feeding.

Regarding the apparent disconnect between the behavioral and imaging experiments, we are puzzled by the statement of the reviewer. In our opinion there is a very clear connection between these two types of data. In short, we show that sensillar and taste peg GRNs play specific roles in yeast feeding and that both sensillar and taste peg GRNs labelled by *Ir76b-GAL4* show physiological responses to the taste of yeast, which are modulated by the internal AA state of the animal. Importantly these changes in yeast sensitivity reflect the changes in yeast feeding elicited by the same nutritional manipulations.

1) In Figure 1 and Figure 1—figure supplement 3, authors argue that amino acids are not the sole stimuli mediating yeast appetite. I would be curious to see how the flies will respond to yeast without amino acids. These yeast sources are commercially available.

We agree that this would be an interesting experiment to do. Unfortunately, we are not aware of any available yeast that lacks amino acids. We think that perhaps the reviewer is referring to yeast nitrogen base without amino acids (Sigma #Y0626). This has been previously claimed by Ganguly and colleagues to be yeast extract without amino acids. However, as stated on the website of Sigma, this is actually a base medium designed for growing yeast and is unrelated to yeast extract.

In the manuscript, major components of yeast taste (glycerol, carbonation, amino acids, etc.) are shown to be not required for yeast appetite, however authors do not suggest or provide a clue about what yeast component is important for flies to taste yeast. I understand this is hard but at least some suggestions would be good to mention in the Discussion.

We now speculate in the discussion on the nature of the yeast components which could be important for mediating the high phagostimulatory power of yeast. In short, we do not think that there is one component which fulfills that specific role, but rather that a blend of different cues leads to the exceptionally high feeding rate observed in protein-deprived mated females. We address this point in the eighth paragraph of the Discussion, and we have added an additional sentence to this passage to clarify our position. A part of this including the added sentence reads:

“Rather, our results suggest that multiple yeast stimuli must coincide to produce a yeast percept, and that yeast GRNs are likely to be specialized to respond to a variety of chemicals normally found in this microorganism. These chemicals are likely to include both those we showed to activate subsets of *Ir76b*GRNs, and other yeast metabolites currently not known to be detected by *Drosophila*.”

2) In Figure 2, authors use the poxn-GAL80 to suppress the expression in GRNs. Some of the Poxn enhancers have been shown to label central neurons in the fly brain (see The Drosophila Pox neuro gene: control of male courtship behavior and fertility as revealed by a complete dissection of all enhancers, Werner Boll, Markus Noll, Development). It would be good to include the expression pattern of the Poxn-4-14-Gal4 or provide evidence that Poxn-Gal80 only suppresses expression of GRNs. In Figure 2 see GFP expression in IR76b-Gal4 outside of chemosensory regions in the fly brain. This expression is suppressed with Poxn-Gal80. There is a possibility that internal yeast sensing might be involved in yeast feeding, thus it would be good to clarify the expression of the enhancer that is used to generate the Gal80.

We now clearly indicate in the list of lines used in this study (Supplementary file 1) that *Poxn-GAL4* corresponds to *Poxn-Gal4-14*. Indeed, *Poxn-GAL4-14* and, therefore also likely *Poxn-GAL80*, is expressed outside of the gustatory system. This has been nicely documented by Boll and Noll and indeed, when we intersect *Poxn-GAL80* with *Ir76b-GAL4*, the expression of a few central cell bodies disappears. We still think, however, that it is valid to claim that the effect on yeast feeding upon silencing of *Ir76b-GAL4, 1261-GAL4* and *Ir25a-GAL4* is likely due to silencing of gustatory neurons. The main argument is that *Ir25a-GAL4* drives expression exclusively in sensory neurons. Given that the effect of silencing with *Ir25a-GAL4* is abrogated when we introduce either *Ir76b-GAL80* or *Poxn-GAL80*, the phenotype of the lines we identified is very likely due to a common population of taste receptor neurons labelled by *1261GAL4, Ir76b-GAL4* and *Ir25a-GAL4*. It is important to note that we are not claiming that there is no postingestive assessment of yeast in the fly; such an effect could additionally contribute to yeast feeding.

Also in Figure 2, there are still GFP positive neurons in VNC. What is the evidence that these neurons are not involved in yeast preference?

In Figure 2 we show the evidence that intersecting *tsh-GAL80* with *Ir76b-GAL4* abolishes expression in the leg chemosensory neurons but does not abolish the silencing effect on yeast feeding. We therefore conclude that silencing of leg chemosensory neurons does not explain the yeast feeding phenotype observed when silencing *Ir76b-GAL4*. Indeed in this intersection experiment, a few cell bodies are still labeled in the VNC. However, given that *Ir25a-GAL4* does not label these neurons (as seen in Figure 1—figure supplement 2 it only labels chemosensory neurons), and the silencing effect of this line is due to the joint population with *Ir76bGAL80*, these neurons are unlikely to contribute to the phenotype.

The authors claim that yeast preference is due to only yeast taste and this sentence is not completely accurate.

We are sorry if we gave the impression that this is the case. We have tried to be as careful as possible to not claim in our manuscript that yeast feeding depends only on taste. In our opinion we are always clear to say that taste plays an important role in mediating yeast feeding. If the reviewer can point us to a specific sentence we would be very happy to edit it to clarify our position.

3) In Figure 2—figure supplement 3 see a lot of variation in different Gr and IR silencing experiments, some of which are significant. Could authors comment on this? Especially Ir60b-Gal4 and Ir94f-Gal4 seem to have an impact on yeast preference. There is a clear difference with Ir94f-Gal4>shi^ts^ at 30 °C vs. 18 °C. However, because authors do not show how controls behave in 18 °C, it is inconclusive. Why are authors not mentioning these results in the text? These two Gal4s have been shown to regulate sugar ingestion and might be involved in yeast appetite too (see A receptor and neuron that activate a circuit limiting sucrose consumption, Joseph R et al. eLife, 2017). This paper also has not been cited in the references.

As stated above, in the original version of the figure we had omitted all the additional control experiments we had done in order to avoid overcrowding the figure. We now show all controls, and when we compare to the genetic background control experiments these two lines do not show a significantly different level of yeast feeding. We have now modified the figure to include all control experiments and hope that the argument for a lack of phenotype is now made more clearly. Furthermore, we agree with the reviewer that Joseph et al., 2017 is a very important and relevant paper, and as such we have added citations to this paper: we now mention that “pharyngeal GRNs are important for […] limiting sucrose intake (Joseph et al., 2017)”; and, we cite this paper when mentioning the lines we chose to manipulate pharyngeal GRNs (subsection “Proboscis gustatory receptor neurons in distinct locations act in parallel to support total yeast feeding”, fifth paragraph).

4) The authors test AMS1 and PMS4 calcium response to carbonation, glycerol, and putrescine, but did not test mixed amino acids, a critical component of the yeast diet (Figure 3—figure supplement 1).

We now include imaging data for the amino acid mix in Figure 4—figure supplement 1. This mix induces very little change in calcium level in the tested neurons.